METHODS

# Homeodynamic feedback inhibition control in whole-brain simulations

**Jan Stasinski** [1,2,3,4], **Halgurd Taher** [1,2], **Jil Mona Meier** [1,2], **Michael Schirner** [1,2,3,4,5], **Dionysios Perdikis** [1,2], **Petra Ritter** [1,2,3,4,5] *

**1** Berlin Institute of Health at Charité, Universitätsmedizin Berlin, Berlin, Germany, **2** Brain Simulation Section, Department of Neurology with Experimental Neurology, Charité, Universitätsmedizin Berlin, Corporate member of Freie Universität Berlin and Humboldt Universität zu Berlin, Berlin, Germany, **3** Bernstein Focus State Dependencies of Learning and Bernstein Center for Computational Neuroscience, Berlin, Germany, **4** Einstein Center for Neuroscience Berlin, Berlin, Germany, **5** Einstein Center Digital Future, Berlin, Germany

* petra.ritter@bih-charite.de

**Data Availability Statement:** All data used in this study was derived from the Human Connectome Project Young Adult study available in the repository https://www.humanconnectome.org/

## Abstract

Simulations of large-scale brain dynamics are often impacted by overexcitation resulting from heavy-tailed structural network distributions, leading to biologically implausible simulation results. We implement a homeodynamic plasticity mechanism, known from other modeling work, in the widely used Jansen-Rit neural mass model for The Virtual Brain (TVB) simulation framework. We aim at heterogeneously adjusting the inhibitory coupling weights to reach desired dynamic regimes in each brain region. We show that, by using this dynamic approach, we can control the target activity level to obtain biologically plausible brain simulations, including post-synaptic potentials and blood-oxygen-level-dependent functional magnetic resonance imaging (fMRI) activity. We demonstrate that the derived dynamic Feedback Inhibitory Control (dFIC) can be used to enable increased variability of model dynamics. We derive the conditions under which the simulated brain activity converges to a predefined target level analytically and via simulations. We highlight the benefits of dFIC in the context of fitting the TVB model to static and dynamic measures of fMRI empirical data, accounting for global synchronization across the whole brain. The proposed novel method helps computational neuroscientists, especially TVB users, to easily "tune" brain models to desired dynamical regimes depending on the specific requirements of each study. The presented method is a steppingstone towards increased biological realism in brain network models and a valuable tool to better understand their underlying behavior.

## Author summary

We introduce the dynamic inhibitory plasticity mechanism (dFIC) in the widely used Jansen-Rit brain network model. The mechanism allows for adapting inhibitory coupling weights based on a synaptic plasticity-inspired rule. Our method effectively balances long-range excitation and local feedback inhibition, allowing better control over the brain network model's dynamics and analysis of the tuning process. We study the conditions,

study/hcp-young-adult. The derived data generated in this study are available under restricted access due to EU data privacy laws, access can be obtained within a timeframe of one month from the corresponding author Petra Ritter as processing and sharing is subject to the European Union General Data Protection Regulation (GDPR), requiring a written data processing agreement, involving the relevant local data protection authorities, for compliance with the standard contractual clauses by the European Commission for the processing of personal data under GDPR (https://commission.europa.eu/publications/standard-contractual-clauses-controllers-and-processors-eueea_en). All custom codes used in this study are freely available at GitHub (https://github.com/BrainModes/dFIC_code/)83 licensed under the EUPL-1.2-or-later. Custom codes were implemented using Python version 3.11.5 and packages in conda 23.11.0.

**Funding:** This work was supported by the Virtual Research Environment at the Charité Berlin – a node of EBRAINS Health Data Cloud. Part of computation has been performed on the HPC for Research cluster of the Berlin Institute of Health. PR was the recipient of all the funding below. PR acknowledges support by EU Horizon Europe program Horizon EBRAINS2.0 (101147319), Virtual Brain Twin (101137289), EBRAINS-PREP (101079717), AISN – (101057655), EBRAIN-Health (101058516), Digital Europe TEF-Health (101100700), EU H2020 Virtual Brain Cloud (826421), Human Brain Project SGA2 (785907); Human Brain Project SGA3 (945539), ERC Consolidator (683049); German Research Foundation SFB 1436 (425899996); SFB 1315 (327654276); SFB 936 (178316478); SFB-TRR 295 (424778381); SPP Computational Connectomics (RI 2073/6-1), (RI 2073/10-2), (RI 2073/9-1); DFG Clinical Research Group BECAUSE-Y (504745852), PHRASE Horizon EIC grant (101058240); Berlin Institute of Health & Foundation Charité, Johanna Quandt Excellence Initiative; ERAPerMed Pattern-Cog (2522FSB904). The funders had no role in study design, data collection and analysis, decision to publish, or preparation of the manuscript.

**Competing interests:** The authors have declared that no competing interests exist.

boundaries, and consequences of using the proposed method on different scales and modalities. We demonstrate that under certain conditions dFIC leads to improved variability of behavior, more biologically plausible simulation results and better fits to empirical data. Our solution is presented as an effective method for improving the fitting simulated to empirical data, by allowing computational neuroscientists to set the activity according to specific study requirements.

## Introduction

Brain network modeling (BNM) allows us to investigate how phenomena observed in electrophysiological and neuroimaging experiments relate to underlying biology, without the necessity of invasive and resource-consuming experiments. Mean-field simulations of whole-brain dynamics enable formulating and testing hypotheses about the underlying mechanisms mainly at mesoscopic and macroscopic scales, while at the same time considering individual differences in brain anatomy, including the roles of subunits such as specific cell types, connections, or synaptic mechanisms. The Virtual Brain (TVB; thevirtualbrain.org) offers a multifunctional platform for running individualized brain simulations using BNMs [1–3]. In the TVB framework, simulations are designed by coupling neural mass models, each corresponding to a brain region, i.e., a node, in the structural brain network. The edges of the structural brain network consist of white-matter tracts and are approximated from diffusion-weighted Magnetic Resonance Imaging (MRI) data using tractography [4], resulting in weights (structural connectivity (SC)) and tract lengths matrices. Typically, the output of BNM simulations is fitted to empirical data to explain high-level properties of empirical signals, including functional magnetic resonance imaging (fMRI) networks, electroencephalography (EEG), or firing rates of neuronal populations [5–8]. In recent years, this approach has found its use in many theoretical and clinical applications. Whole-brain simulations were used to investigate brain dynamics in Alzheimer's disease [9–11], epilepsy [12,13] and schizophrenia [14]. Theoretical applications include studies of excitatory-inhibitory balance and emergent neural mechanisms over a range of temporal and spatial scales [15–17].

One challenge of large-scale brain simulations is to obtain 'biologically plausible' brain signals that represent aspects of real brain phenomena, such as network interactions and dynamic changes due to interventions or pathology. Here we describe brain signals as 'biologically plausible' if they resemble characteristics of empirical data, e.g., firing rates, synchronicity, or functional connectivity (FC). Experimental studies have shown that neuronal populations in resting state exhibit on average low to moderate levels of asynchronous activity, with low variability between regions. Unlike the saturation of activity or the large spread of firing rates that can be obtained by a connectome that spans several orders of magnitude [18–21]. For example, *in vivo* studies report a $\sim 3$ Hz firing rate for excitatory neurons and irregular (Poisson-like) cortical activity [22,23]. Thus, simulations should produce similar characteristics. For this purpose, an established method is to fit the global coupling parameter to adjust the input variability across regions. However, global coupling has two distinct effects: (i) it controls interactions and therefore correlations among regions and (ii) at the same time impacts their excitability. On the one hand, lowering global coupling to avoid overexcitation can result in decreased network interactions, potentially rendering them incomparable to empirical data. On the other hand, overexcitation and highly correlated activity driven by strong coupling can be considered implausible and unlikely to exhibit switching between different regimes as some of them require small inputs and low levels of activity to be present.

Additionally, we argue that for the simulated signals the considered 'biologically plausible' they should exhibit complex dynamics like criticality and multistability, which have been discussed as important for brain dynamics [24–27], both in resting state and in the context of the brain function. Previous research suggests their role in working memory, decision making [28,29], and processing of information [30–32] as they are often associated with a balance between global integration and local segregation in brain networks [33–36]. Complex behavior in dynamical systems corresponds to e.g. (i) multistable dynamics, which in the presence of varying input can lead to switching between different stable attractors, (ii) metastable dynamics (slowing down around attractor "ghosts" [30], before switching to other attractors' neighborhoods in state space), or (iii) chaos, i.e., unstable but bounded dynamics. The presence of these complex dynamics hinges on a calibration of the model towards critical points, i.e., bifurcations or multistable regimes, where local input perturbations can cause a qualitative change in the model's dynamics [37,38]. However, in the context of BNMs, the coupling weights are often characterized by heavy-tailed distributions [39–42] which leads to a high variability across regions in terms of their external inputs in a BNM. Therefore, controlling the dynamics of a BNM towards criticality or multistability is non-trivial, while being crucial for fitting procedures and model calibration.

Experiments conducted *in vivo*, *in vitro*, and *in silico* have demonstrated that steering the system toward or away from criticality can be achieved by dynamic adjustments of the excitation-inhibition (E-I) balance [43–45]. From a modeling perspective, two closely related plasticity mechanisms have been proposed: non-local E-I balancing [46] and local feedback inhibition control (FIC) [16]. At its core, FIC is a mechanism to locally regulate inhibition in mean-field models [47] and enforce a target firing rate. Schirner and colleagues successfully used such synaptic plasticity-like learning rule, which dynamically adjusts E-I balance between proximal nodes in the context of decision making [46]. In the presence of FIC, this dynamic balancing can control synchrony between pairs of regions, to explicitly reproduce empirical FC patterns in the context of fluid-intelligence performance. Similar work was done with the Wilson-Cowan model to steer the system toward criticality [48–51].

Our study implements a homeodynamic plasticity mechanism (dFIC) in the Jansen-Rit (JR) neural mass model [52]. This versatile model generates slow- and fast-periodic oscillations in the δ- and α-bands, making it popular for studying network dynamics and fitting simulated time-series to empirical data including EEG, MEG, and resting-state fMRI signals [7,9,14,53–58]. Multistability is an inherent property of this model, as two fixed points (FPs), two limit cycles (LCs), or a FP and a LC, can coexist for specific parameter configurations [53]. We consider this property to be relevant for simulating blood-oxygen-level-dependent (BOLD) fMRI signals. In particular, BOLD FC dynamics (FCD) are characterized by the reoccurrence of FC networks (e.g., default-mode network) over time windows of tens of seconds, as opposed to the fast dynamics of many neural mass models which occur on a millisecond timescale [29]. Occasional, noise-induced switching between different attractors of these fast systems can give rise to the additional slower timescales observed in FCD. Our aim is to promote the above-mentioned complex dynamics through FIC. So far, the only known implementation of FIC in the JR model was proposed in the work of Coronel-Oliveros et al. [57], where a similar mechanism was used to constrain the activity of the JR model in the context of Alzheimer's disease.

In this work, we study the function and limitations of dFIC in the JR model. Our motivation is two-fold. First, to counteract heavy-tailed SC matrices and to compensate the over-excitation due to long-range coupling. This can be seen as decorrelation of large-scale network effects and local activity of BNMs. Second, to tune the activity and the range of traversed dynamical regimes of JR nodes in a network, regardless of the chosen method of structural connectivity normalization. In other words, long-range coupling and its strength can be

chosen without impairing criticality and multistability, as well as without violating empirical constraints.

The structure of the article is as follows: we first provide an analysis of the dFIC mechanism in the uncoupled JR model. Further, we demonstrate the dFIC mechanism in a small network of 4 nodes before moving to a whole-brain network using the average connectome from the Human Connectome Project Young Adult data set [58,59]. We analyze how the global dynamics of a large-scale brain network model are affected by locally regulating feedback inhibition. Through stochastic simulations and analytical methods, we highlight the impact of using dFIC focusing on specific activity targets and their consequences in parameter estimation procedures in a whole-brain simulation paradigm. We demonstrate the benefits of applying dFIC with a specific focus on increased switching between models' attractors, both on the single-node and whole-network level. Finally, we discuss the process of fitting the simulated to empirical data using a custom fitness function, accounting for static and dynamical FC as well as for the overall level of synchronization in BOLD signals.

## Methods

The following section outlines the dFIC mechanism and provides detailed descriptions of steps taken to introduce and set up dFIC in the JR model. First, we introduce the equations, state variables and parameters that were added to the original JR model and outline the workflow used to obtain the simulated data. Next, we describe the structural connectivity matrices used as a basis for our simulations. Finally, we list the used measures and statistical tools to describe and quantify the results.

### Equations of the JR model

The JR model represents a local cortical circuit of three interconnected populations captured by six differential equations, two for pyramidal projection neurons **Eqs (1A–1B)**, two for excitatory **Eqs (1C–1D)** and two for inhibitory interneurons **Eqs (1E–1F)**, which form feedback loops. Neural populations are characterized by a second-order differential operator that converts the average incoming spike rate into the average membrane potential, whereas a non-linear, sigmoidal, function is applied to transform the average membrane potential into the average output spike rate. The state variables $y_0$, $y_1$ and $y_2$ describe the outputs of pyramidal, excitatory, and inhibitory postsynaptic potential blocks (EPSP and IPSP) respectively. The postsynaptic potential (PSP) of the $i$-th pyramidal population is given by the difference $PSP_i = y_{1,i} - y_{2,i}$. We implemented one additional state variable to the tuning model, namely $wFIC_i$, which describes the adaptive strength of the inhibitory input onto the pyramidal cells of region $i$ in **Eq (1B)**. The graphical representation of the model can be found in the **Fig 1**.

In the following equations, $C$ denotes the SC matrix and $C_{ij}$ are the entries of that matrix, i.e., the connection weights between regions $i$ and $j$. The symbol $N_{ROIs} = 84$ denotes the number of nodes in the connectome. Analogously, $\theta_{ij}$ is the time delay and is computed from the tract lengths between regions $i$ and $j$, as well as the conduction speed value, which was set at 5 m/s. Additionally, we introduce $I_{ext,i}$—the total external current received by region $i$, consisting of a constant component $\mu$ and a term representing long-range excitatory coupling, which sums over the outgoing firing rates of all regions $j$, weighted by the corresponding SC element $C_{ij}$ **Eq (3)**. The parameter values used for simulations are listed in **Table 1** and based on physiological considerations in the original Jansen and Rit publication [52]. The amplitudes of the EPSP and IPSP are determined by $A = 3.25$ mV and $B = 22$ mV for the excitatory and inhibitory cells, respectively. The corresponding durations of EPSP and IPSP are inversely proportional to $a = 0.1$ ms$^{-1}$ and $b = 0.05$ ms$^{-1}$. As in the original work [52], the firing threshold is

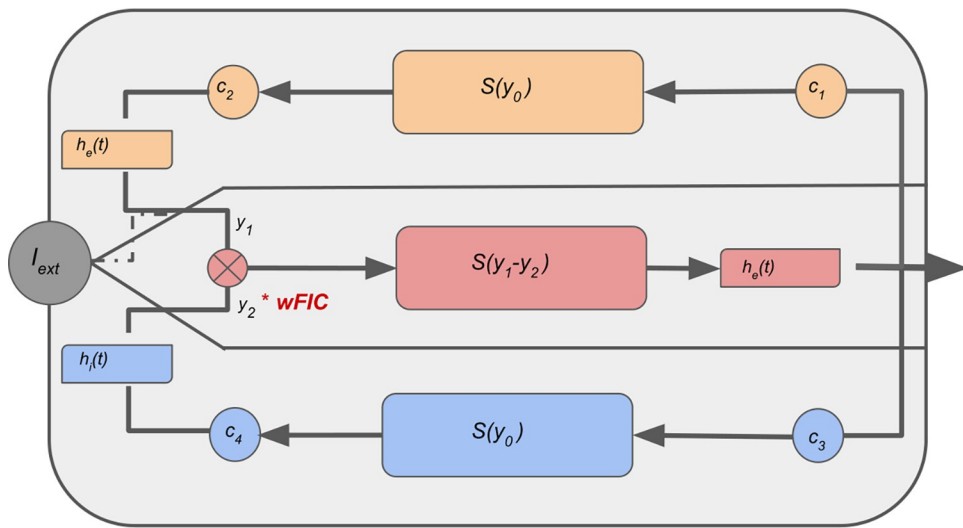

**Fig 1. The graphical representation of the JR model with the introduced heterogenous inhibitory scaling variable** **wFIC_i.** The three populations Excitatory Interneurons, Pyramidal Cells and Inhibitory Interneurons are marked in orange, red and blue respectively. The sigmoidal is denoted with S and $h_e(t)$ and $h_i(t)$ represent the impulse response resulting from the model equations **Eqs (1A–1F)**.

given by $v_0 = 6$ mV. The remaining parameters of the sigmoidal transfer function in **Eq (2)** were set at $v_{max} = 0.0025$ s$^{-1}$, $e_0 = 2.5$ s$^{-1}$ and $r = 0.56$ mV$^{-1}$. The constant $J = 135$ corresponds to the number of the synaptic contacts, which is scaled for individual connections with $c_1$, $c_2$, $c_3$ and $c_4$.

$$\dot{y}_{0,i}(t) = y_{3,i}(t)$$

$$\dot{y}_{3,i}(t) = AaS[y_{1,i}(t) - wFICi(t) \cdot y_{2,i}(t)] - 2ay_{3,i}(t) - a^2 y_{0,i}(t)$$

$$\dot{y}_{1,i}(t) = y_{4,i}(t)$$

**Table 1. Parameters and their values.** These values are fixed throughout this work, if not stated differently. EPSP: excitatory post-synaptic potential, IPSP: inhibitory post-synaptic potential, PC: pyramidal cells, EI: excitatory interneurons, II: inhibitory interneurons.

| Symbol | Description | Value |
|---|---|---|
| $A$ | Maximum amplitude EPSP | 3.25 mV |
| $B$ | Maximum amplitude of IPSP | 22 mV |
| $a$ | Reciprocal of excitatory time constant | 0.1 ms$^{-1}$ |
| $b$ | Reciprocal of inhibitory time constant | 0.05 ms$^{-1}$ |
| $v_0$ | Firing threshold | 6 mV |
| $r$ | Steepness of sigmoidal | 0.56 mV$^{-1}$ |
| $c_1$ | Local coupling from PC to EI | $1 \cdot 135$ |
| $c_2$ | Local coupling from EI to PC | $0.8 \cdot 135$ |
| $c_3$ | Local coupling from PC to II | $0.25 \cdot 135$ |
| $c_4$ | Local coupling from II to PC | $0.25 \cdot 135$ |
| $v_{max}$ | Maximum firing rate of neural population | 0.0025 s$^{-1}$ |
| $\eta$ | Adaptation rate | $0.005 \cdot$ 1/ms·(mV)$^2$ |
| $\tau_d$ | Adaptation window | 1000 ms |

$$\dot{y}_{4,i}(t) = Aa[c_2 S(c_1 y_{0,i}(t)) + I_{\text{ext},i}(t)] - 2a y_{4,i}(t) - a^2 y_{1,i}(t)$$

$$\dot{y}_{2,i}(t) = y_{5,i}(t)$$

$$\dot{y}_{5,i}(t) = Bb c_4 S[c_3 y_{0,i}(t)] - 2b y_{5,i}(t) - b^2 y_{2,i}(t) \tag{1A–1F}$$

$$S(v) = \frac{2v_{max}}{1 + exp(r(v_0 - v))} \tag{2}$$

$$I_{\text{ext},i}(t) = \mu + G \sum_{j=1}^{N_{\text{ROIs}}} C_{ij} S[y_{1,j}(t - \theta_{ij}) - wFIC_j(t) y_{2,j}(t - \theta_{ij})] \tag{3}$$

## Regimes of the JR model

The JR model exhibits a complex repertoire of regimes and bifurcation points that is dependent on its parameters depicted in **Fig 2**. The presence of bistability, where two stable states coexist within a specific parameter range, is one of the fundamental features of this model. In the case of the uncoupled JR model one of the regimes generates the α-band (8–12 Hz) oscillatory activity–here referred to as fast LCs with frequencies in the range 10–12 Hz. Another regime gives rise to slower, δ-band (1–4 Hz) and θ-band (4–5 Hz) rhythm oscillations—referred to as slow LCs. In the following we will omit the region index $i$ for the case $G = 0$, corresponding to an isolated JR model. Given a deterministic setting, as the input $\mu$ is increased

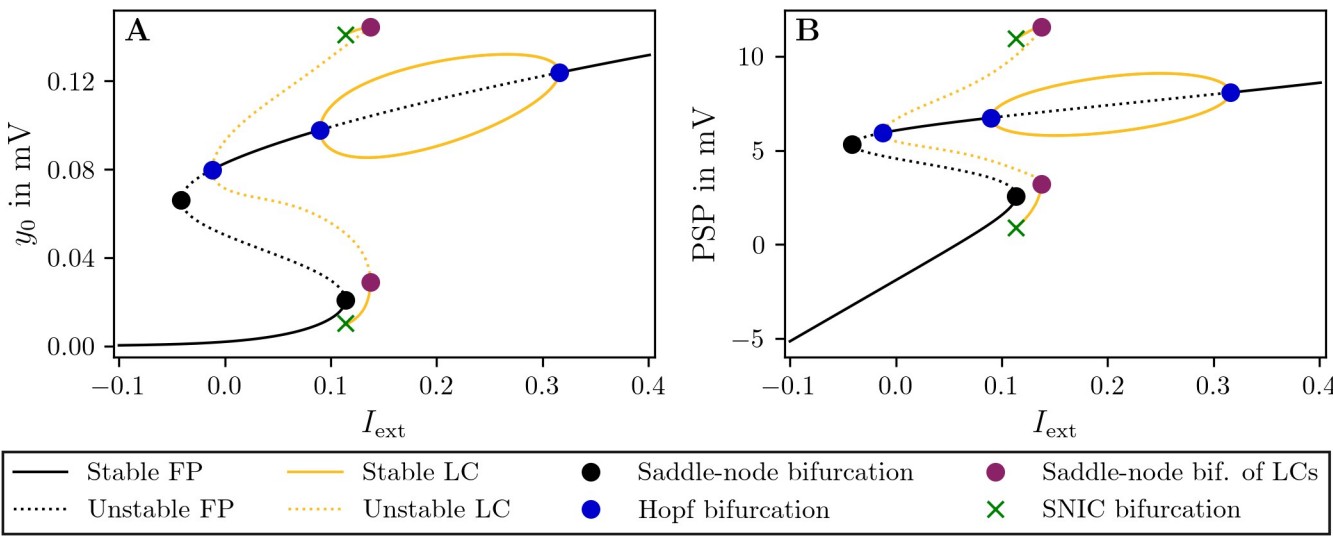

**Fig 2. Bifurcation diagrams of a single uncoupled JR model. A:** State variable $y_0$ of fixed points (FPs) and limit cycles (LCs) vs. constant external input $I_{\text{ext}} = \mu = const.$ At low inputs $I_{\text{ext}} \approx -0.1$, one can find a branch of stable FPs (black line) which undergoes a series of bifurcations. The branch folds at $I_{\text{ext}} \approx -0.1$, where a saddle-node bifurcation (SN, black dot) destabilizes it (dashed black line). A second SN follows, without changing the stability. The two SN result in an S-shaped bifurcation structure. The upper branch stabilizes via a subcritical Hopf bifurcation near $I_{\text{ext}} \approx 0$, which gives rise to unstable LCs (dashed orange line). The LC branch becomes stable (solid orange line) through a SN of LCs (purple dot) and terminates through a saddle-node on invariant circle bifurcation (SNIC, green cross). The upper stable FP branch destabilizes and stabilizes again through two supercritical Hopf bifurcations, between which one can find another family of LCs. **B:** Same bifurcation structure showing the postsynaptic potential PSP vs. $I_{\text{ext}}$. The bifurcation diagram was computed numerically using AUTO-07p [60]. The parameter values are given in **Table 1**.

from $I_{ext} = 0$ to $I_{ext} < 0.12$, the postsynaptic potential increases steadily, and the model's behavior is determined fixed points (non-oscillatory regime). This state configuration of the system can be described as sub-bistable and corresponds to $0.007 < y_0 < 0.019$. Only when input to an individual node reaches $I_{ext} > 0.12$, the system starts exhibiting bistability between different regimes. The first bistability exists between the low- and high-activity fixed points (FPs) for $I_{ext} \in [-0.012, 0.09]$, the second one between the low activity FPs and the fast LC for $I_{ext} \in [0.09, 0.113]$, and the third one between the fast and slow LCs for $I_{ext} \in [0.113, 0.137]$. Further increases to the input, $I_{ext} > 0.137$, place the system in the fast LC and $I_{ext} > 0.31$ leads to a high-activity FP (non-oscillatory regime), which we jointly refer to as super-bistable regime, where $y_0 > 0.1$. Importantly, when coupling multiple JR nodes, the network effects can give rise to dynamics not captured by the bifurcation diagram of a single, uncoupled node. Nevertheless, the bifurcation diagram for the uncoupled node can approximate network effects under the assumption that each node receives a similar slowly varying "mean-field" input with small noise fluctuations, Therefore, the bifurcation diagram of a single node only approximates the behavior of a network node and may not be precise.

## Dynamic feedback inhibitory control

Importantly, due to recurrent activation in the network case, the overall sum of incoming activity to a network node can increase significantly compared to the single-node scenario. The increase in the input to multiple single nodes results in an increase in the overall excitation in the system and an effectively reduced repertoire of available regimes that the individual nodes can achieve. Such a system exhibits behaviors associated with high-activity states, making it impossible to reach the regimes requiring weaker inputs. In the case of JR, such configuration places most nodes in the fast LC or, in extreme cases, near the high FP and leads to increased synchronization of the activity across nodes. Another consequence is the decreased variability of derived slow timescale signals such as BOLD, since they are computed from PSPs of low variability. dFIC was conceived as a solution that allows counteracting over-excitatory network effects by locally adjusting the inhibitory strength of nodes to reflect the overall strength of incoming connections from other nodes such that the long-term average output PSP activity of pyramidal cells of each node is fixed at a chosen level.

Our solution splits the simulation process into two parts: the deterministic tuning process and the stochastic post-FIC simulations. In the first step, we implement the tuning process, which dynamically adjusts the weights of the $wFIC_i$ state variables to bring the model's activity to the desired level. To achieve this, we add two additional, activity-detection equations **Eqs (4A–4B)** and one equation describing the evolution of the $wFIC_i$ variables **Eq (4C)**, based on a gradient-descent learning rule proposed by Vogels [61], first implemented in large-scale whole-brain networks by Schirner et al. [17]. The activity-detection equations are used to extract a slowly varying average of the two state variables necessary for computing the learning process during the tuning step: $y_{0,i}^d$ and $y_{2,i}^d$, for averaged PSP output of the pyramidal and inhibitory populations, respectively. Thus, fast oscillations and short-term variability of $y_{0,i}$ and $y_{2,i}$ are averaged out. We chose to include such equations to facilitate our analytics by making a fast- and slow-time-scale separation more explicit, as well as in order to disassociate the slow averaging dynamics from the particular form of the learning dynamics, and given that some slow averaging is in all cases necessary to filter out fast noisy fluctuations (in the case of FP targets) or oscillations (in the case of LC). The **Eq (4A)** provides a time-averaged slow detection of the ongoing firing rate as fitting target, which ensures that $wFIC_i$ variable does not

fluctuate or oscillate.

$$\dot{y}_{0,i}^d(t) = \frac{1}{\tau_d}\left(y_{0,i} - y_{0,i}^d\right)$$

$$\dot{y}_{2,i}^d(t) = \frac{1}{\tau_d}\left(y_{2,i} - y_{2,i}^d\right)$$

$$w\dot{F}ICi(t) = \eta y_{2,i}^d(t)(y_{0,i}^d(t) - y_0^{\text{target}}) \qquad (4A\text{–}4C)$$

The crucial parameter defining the target of the dFIC procedure is $y_0^{\text{target}}$, which is the desired value to which the long-term (i.e., at a slow time scale) average activity of the system is being brought to by the tuning process. This value must correspond to one of the attractors (FP or LC) of the uncoupled, original JR model (**Fig 2**). For point attractors, the value corresponds to their exact location, whereas for LCs the value corresponds to their centers (in practice approximated as the average activity value over several cycles). The $\eta$ parameter describes the learning rate at which the $wFIC_i$ variable is being adjusted during tuning to reach $y_0^{\text{target}}$. The time constant of the detection equations is denoted by $\tau_d$. The parameters $\eta$ and $\tau_d$ can be adjusted depending on the desired target dynamical regime of the model and simulation length. For example, $\eta$ determines how quickly the gradient in the dFIC algorithm descends. If $\eta$ is chosen too large, the values of the state variables $wFIC_i$ may oscillate. Once the values of $wFIC_i$ have converged, the tuning is completed and post-FIC simulations can be performed, for which $wFIC_i$ is replaced by a constant parameter $pFIC_i$ **Eq (5A)**.

$pFICi = \frac{1}{L}\sum_{t_k < T_{tune} - L}^{T_{tune}} wFICi(t_k)$, where $t_k = kdt$ is the $k$-th timepoint of the numerical simulation.

$$\dot{y}_{3,i}(t) = AaS[y_{1,i}(t) - pFICiy_{2,i}(t)] - 2ay_{3,i}(t) - a^2 y_{0,i}(t) + \sigma \cdot \text{noisei}(t) \qquad (5A\text{–}5B)$$

To obtain $pFIC_i$, we time-averaged the last $L$ = 3000 ms of the total tuning simulation of $T_{tune}$ milliseconds of the $wFIC_i$ state variable from the tuning step. The heterogeneous $pFIC_i$ scales the strength of the inhibitory connection for post-FIC simulations. The graphical representation of the tuning process can be found in **S2 Fig**.

## Simulations

To test the range of values and limits of the parameters of the tuning approach, we ran a series of simulations using a small-network connectome, consisting of 4 nodes with random weights and distances. We used the deterministic Heun's integration scheme for all the combinations of the detection time constant $\tau_d$ and the learning rate $\eta$ with the tuning simulation length $T_{tune}$ = 250 seconds. This analysis constitutes an intermediate step between the analytical single-node analysis and whole-brain simulations, designed to help determine the values of the two tuning parameters $\tau_d$ and $\eta$, and to estimate the necessary length of upcoming whole-brain simulations. In every tuning setup, a 15 s transient has been discarded before the tuning started. We have selected two $y_0^{\text{target}}$ values, namely, $y_0^{\text{target}} = 0.01$ and $y_0^{\text{target}} = 0.103$, to tune the system to, each chosen at one of the attractors of the single-node, original Jansen-Rit model, to maintain the system's dynamics in the proximity of that attractor. For both $y_0^{\text{target}}$ values, we have adjusted the external input value $\mu$ and all of the respective initial conditions, based on the JR model bifurcation diagram of the single node (**Fig 2**), to allow for faster and smoother tuning toward the desired regime: first, at the low stable FP, close to the bifurcation point towards the slow stable LC at $y_0^{\text{target}} = 0.01$, in which case we set the $\mu = 0.09$ and the

randomized initial conditions below the target value, such that the tuning procedure increases the activity of the system to the desired level; second, at the fast, stable LC, close to the bifurcation point towards the slow stable LC at $y_0^{\text{target}} = 0.103$, in which case we set the value of $\mu = 0.14$ and the randomized initial conditions above the target value, so that the tuning procedure decreases the activity of the system to the desired level. We ran small-network simulations for all combinations of the following settings:

- $y_0^{\text{target}} \in \{0.01, 0.103\}$

- $\tau_d \in \{10, 100, 550, 1000\}\ ms$

- $\eta \in \{0.00035, 0.001, 0.005, 0.01\}\ \frac{1}{\text{ms} \cdot (\text{mV})^2}$

- $G \in \{0, 1, 10, 25, 100\}$

Next, to demonstrate the scalability of our solution, we proceed to whole-brain network simulations. We have used the publicly available HCP Young Adult data set, which includes 3T MR imaging data from healthy adult participants (age range 22–35 years) [62]. The average connectome was derived from multimodal neuroimaging data (diffusion-weighted MRI) and T1-weighted MRI using an automatic preprocessing pipeline [59] processed with Desikan-Killiany parcellation [63], composed of 68 cortical gyrally-based regions, and 16 subcortical regions, added according to the subcortical segmentation present in the Freesurfer software [64].

To test the FC fit for different activity targets and corresponding optimal global coupling values, we ran the full two-parameter search for seven low-activity $y_0^{\text{target}} \in \{0.001, 0.004, 0.007, 0.01, 0.013, 0.016, 0.0189\}$, five high-activity $y_0^{\text{target}} \in \{0.1, 0.11, 0.12, 0.13, 0.14\}$ and 30 values of $G \in [0, 30]$, with step of 1, resulting in a total of (5 +7)×30 = 360 parameter combinations. All tuning simulations were set up with a deterministic Heun's integration scheme ($T_{tune}$ = 4 min, timestep $dt$ = 1 ms) and post-FIC simulations were run using Heun's stochastic integration scheme ($T_p$ = 3 min, timestep $dt$ = 1 ms) with additive noise $\sigma = 10^{-7}$ **Eq (5B)**. For each of the post-FIC simulations, we obtained the PSPs of pyramidal cells, for which we performed power spectrum analysis, Poincaré mapping analysis for each node and computed the resting state Blood Oxygen Level Dependent (BOLD) timeseries ($T_{bold}$ = 30 min, repetition time TR = 720 ms) using hemodynamic response functions with the Balloon Model [65]. To compare post-FIC simulations with the original JR setup, we ran respective simulations for all the $G$ values to obtain the simulation results without the dFIC (later referred to as no-FIC simulations). Values of $pFIC_i$ were set to 1 in the no-FIC simulations, while the remaining parameters, including $noise_i$ and $\mu$, corresponded to their post-FIC counterparts.

The introduction of stochasticity has two main consequences. Due to the nonlinearity of the model, it increases the overall input per node, which, especially in case of high $G$ values, causes the constrained activity of the post-FIC system to deviate from the $y_0^{\text{target}}$ activity level. Additionally, as demonstrated by further results, noise introduces variability to allow the individual nodes to switch dynamical regimes in the proximity of the chosen activity level. Measuring and quantifying switching behavior in neural systems can be done with the help of Poincaré mapping analysis [66,67].

We implemented a method for a Poincaré mapping involving the analysis of a PSP timeseries [68–70]. Poincaré maps are derived from the Poincaré sections, representing a projection of a dynamical system's behavior onto a lower-dimensional space. Here we take the simulated PSP timeseries $PSP_i(t)$ of region $i$ and determine the $k$-th local maximum $PSP_{i,k}$ numerically by comparing each value at a given time point with neighboring points in a 100

ms time window. The recurrence map given by $PSP_{i,k} \rightarrow PSP_{i,k+1}$ yields a point cloud in which each point corresponds to state changes of individual nodes. The method essentially maps the continuous dynamical system into a discrete one, acting like snapshot of changes in the maxima of the signal. Nodes which stay nearby an attractor, be it FP or LC, exhibit fluctuations around that attractor due to noise. Consequently, the points $(PSP_{i,k}, PSP_{i,k+1})$ spread around the attractor–for FP around the PSP value of the FP and for LCs around all PSP value of local maxima within one period (**S1 Fig**).

## Fitting methods

For BOLD, we computed FC matrices using the Pearson correlation coefficient. To assess the fit between empirical and simulated data we have compared our simulation results to both the average FC matrix and to an average functional connectivity dynamics (FCD) distribution derived from the Human Connectome Project Young Adult data set [62]. For each of the parameter combinations, we computed the value of FC of the BOLD signals and calculated the Pearson Correlation between simulated and empirical data ($R_{FC}$). Further we utilized average BOLD FC (the mean of the upper triangle of the empirical FC matrix), as a threshold for the fitting procedure of the simulated FCs to avoid over-synchronization. For our BOLD FC fitting, we have discarded the simulations with $FC_{mean} > 0.25$, slightly above the average of the empirical FC matrix ($FC_{mean} = 0.23$). This process helped us to reject biologically unplausible and over-synchronized FC data (**S4 Fig**). FCDs were computed using multiple FCs per simulation obtained by sliding window of size of 60 seconds and increment with $1\ TR = 720$ ms. The fit was quantified by the Kolmogorov-Smirnov distance ($1-KSD_{FCD}$) between the distributions of empirical FCD values and each of the simulated FCDs. Additionally, we combined the fits of $R_{FC}$, $1-KSD_{FCD}$, and a synchronization threshold into a single multimodal factor (MMF) which is described by **Eq (6)**. The coefficients $d_1$, $d_2$ were set at 1, and 0.75, respectively. The selection of coefficients was motivated by the general idea that both $R_{FC}$, and $1-KSD_{FCD}$ should contribute to the final measure equally despite having different maximum values, with 0.6 and 0.8 respectively. The coefficient $d_2$ was set at 0.75 to fulfil this requirement. We found this combined method useful for assessing the biological plausibility of our simulated data, but it is worth noting that the threshold value and coefficients are likely dataset specific and should not be used without numerical justification and testing.

$$MMF = \frac{(d_1 R_{FC}) + (d_2(1 - KSD_{FCD}))}{(d_1 + d_2)} \tag{6}$$

To confirm that no additional simulations are needed for a robust estimation of the MMF measure, we performed a bootstrapping statistical analysis, for each parameter combination ($G$, $y_0^{target}$) and for each condition (post-FIC and no-FIC simulations). To that end, we generated a bootstrapping sample distribution by resampling with replacement from an original set of–per parameter combination–MMF values, computed for overlapping windows of a shorter time length (19.2 min that we extracted from the original 30 min simulations. The results show that the MMF values we computed were robust already across such shorter time windows, for more details see **S1 Appendix**.

## Results

### Control of a single Jansen-Rit population

In this section we present results on the control of a single JR node in presence of the homeodynamic inhibitory plasticity mechanism dFIC. We outline how dFIC achieves a desired

activity level when the external input $I_{ext}$ = const., since this is a prerequisite for the functioning of dFIC in a network setup, where $I_{ext}$ is time dependent. The following semi-analytical methodology is based on numerical bifurcation analysis and slow-fast dissection [71,72]. To start with, we note that the full system for an isolated node, described in **Eqs (1),(2)** and **Eq (4)**, can possess different timescales, depending on the parameter values. The JR dynamics are governed by the time constants $1/a$ = 10 ms and $1/b$ = 20 ms. Typical values for the detection time constant $\tau_d$ and the learning rate $\eta$ used in this work are $\tau_d$ = 1000 ms and $1/\eta$ = 400 (mV)$^2$ ms. With this configuration the full system has three timescales. For simplicity however, we assume a two-timescale system, by posing that $\tau_d$ and $1/\eta$ are large enough, such that (i) $y_0^d$ and $y_2^d$ converge to long-term temporal averages and (ii) changes in $wFIC$ are slow, giving time to the JR state variables to converge to the proximity of attractors present at constant $wFIC$. This assumption allows us to apply slow-fast dissection, to understand the dynamics of the full system (i.e., single node with dFIC), which emerges from the dynamics of the no-FIC system through a slow variation of the $wFIC$.

In the limit case $1/\tau_d \rightarrow 0$, $\eta \rightarrow 0$, corresponding to no-FIC, $wFIC$ becomes a parameter in the remaining equations **Eqs (1)** and **(2)** $y_0^d$ and $y_2^d$ become the long-term temporal averages of $y_0$ and $y_2$, respectively. To understand the dynamics in this limit case, we average the LCs of the original JR model (**Fig 2**) over one period, as shown in **Fig 3A** for $wFIC$ = 1.

Notably, $wFIC$ = 1 yields the original bifurcation landscape of the JR model, whereas values $wFIC \neq 1$ reshape this structure. To capture this alteration, one can compute $y_0^d$ vs. $I_{ext}$ bifurcation diagrams for an interval of $wFIC$ values and obtain surfaces of FPs and LCs, as shown in **Fig 3B**. Initial conditions and a given point ($I_{ext}$, $wFIC$) in the parameter plane will determine which of the attractors of the JR model the dynamics will approach. At the same time, $y_0^d$ and $y_2^d$ will approach the corresponding temporal average of that attractor. This attractor, be it a FP or a LC, will not satisfy $y_0^d = y_0^{target}$, unless ($I_{ext}$, $wFIC$) is chosen appropriately. In other words, the average activity is usually lower or higher than the desired target.

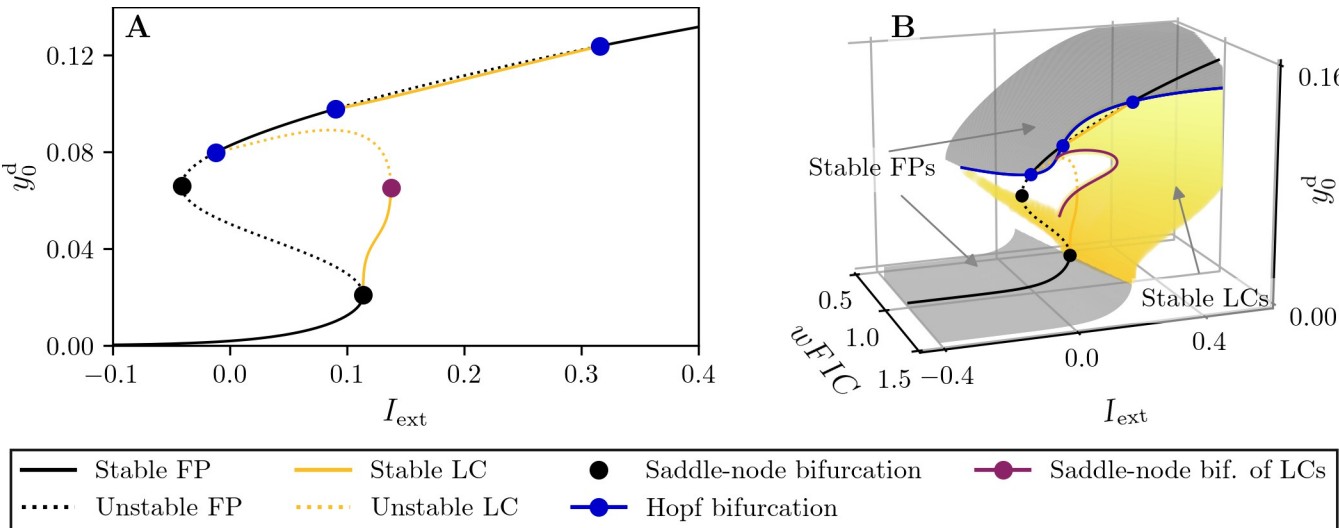

**Fig 3. Bifurcation structure of the uncoupled JR model with averaged limit cycles (LCs). A:** Average $y_0^d$ versus external current $I_{ext}$ for $wFIC$ = 1. The solid (dashed) black lines correspond to families of stable (unstable) fixed points (FPs). For families of stable (unstable) LCs, the average value of $y_0$ over one cycle is computed and depicted as solid (dashed) orange lines. **B:** Bifurcation structure in $wFIC - I_{ext} - y_0^d$ space. The bifurcation structure of panel A is depicted at $wFIC$ = 1. When computed on an interval of $wFIC$, stable branches of this structure form an attracting surface with a lower and an upper sheet corresponding to families of stable FPs (gray surface) and LCs (yellow surface). The upper boundaries of the stable LC surface are given by a family of Hopf bifurcations (blue line) and saddle-node bifurcations of LCs (purple line). The bifurcation diagrams were computed numerically using AUTO-07p [60] and the parameter values given in Table 1.

In the case of a small $\eta > 0$, *wFIC* is a subject to a slow drift, proportional to $y_0^{\mathrm{d}} - y_0^{\mathrm{target}}$. An increase of *wFIC* will cause a decrease in $y_0^{\mathrm{d}}$ and vice versa. Therefore, if $y_0^{\mathrm{d}} > y_0^{\mathrm{target}}$, *wFIC* slowly drift towards larger values, causing a decrease of $y_0^{\mathrm{d}}$, whereas for $y_0^{\mathrm{d}} < y_0^{\mathrm{target}}$, *wFIC* drifts to smaller value, causing an increase of $y_0^{\mathrm{d}}$. Eventually, this leads to $y_0^{\mathrm{d}} = y_0^{\mathrm{target}}$, the drift of *wFIC* stops–it has reached equilibrium. This behavior can be understood analytically. From the representation in **Fig 3B**, one can assume that $y_0^d(wFIC)$ and $y_0^d(wFIC)$ are functions of *wFIC*. For the dFIC equilibrium we follow that:

$$\dot{w}FIC = g(wFIC) := \eta\, y_2^d(wFIC)[y_0^d(wFIC) - y_0^{\mathrm{target}}] \tag{7}$$

$$\dot{w}FIC = 0 \Leftrightarrow y_0^d(wFIC) = y_0^{\mathrm{target}} \tag{8}$$

Note that $y_2^d(wFIC) > 0$ holds in the parameter intervals considered in this work. A linear stability analysis at the above equilibrium yields $\frac{dg}{dwFIC} = \eta y_2^d \frac{dy_0^d}{dwFIC}$. Therefore, if $\frac{dy_0^d}{dwFIC} < 0$ is fulfilled at the dFIC equilibrium, it will be a stable equilibrium. Given the JR model with the implementation of dFIC as an inhibitory plasticity mechanism, this condition—by construction—is always fulfilled. To sum up, a dFIC equilibrium exists at a given $I_{\mathrm{ext}}$ if a stable FP or LC with $y_0^{\mathrm{d}} = y_0^{\mathrm{target}}$ at this exact $I_{\mathrm{ext}}$ exists and it follows automatically that this equilibrium is stable.

In the following, we provide simulation results to support the above analysis, using three representative examples shown in **Fig 4**. They also give insight into the functioning of dFIC and allow to relate *wFIC* dynamics to the JR bifurcation structure. In the first case, shown in **Fig 4A1, 4B1 and 4C1**, we set $I_{\mathrm{ext}} = 0.2$, for which the original JR model only possesses stable LCs with activity levels beyond the desired $y_0^{\mathrm{target}} = 0.01$. Consequently, in the absence of dFIC, the system converges to this LC. The trajectory with active dFIC however, exhibits a drift in *wFIC*, forcing the activity level to the desired target, which is a fixed point of the JR model. The second case in **Fig 4A2, 4B2 and 4C2**, on the other hand aims to reach a LC at a target of $y_0^{\mathrm{target}} = 0.1$. In the absence of dFIC, the system goes beyond this target and converges to a fixed point on the upper FP sheet. With dFIC, the activity is reduced and forced onto a fast LC of the JR model. In the last case, **Fig 4A3, 4B3 and 4C3**, the activity is increased from the fast LC to a larger level, corresponding to a fixed point on the upper sheet.

Having dFIC equilibria over a wide span of $I_{\mathrm{ext}}$ values is desirable: first, the external current is often a free parameter in neural mass models and second, $I_{\mathrm{ext}}$ will be dynamic and subject to fluctuations in a network setup, as we discuss in the next section. Hence, we look for intersections of attractors, depicted in **Figs 3B** and **4** with the plane $y_0^{\mathrm{d}} = y_0^{\mathrm{target}}$. These intersections are shown in **Fig 4** as red lines and represent the set of ($I_{\mathrm{ext}}$, *wFIC*) pairs, for which $y_0^{\mathrm{d}} = y_0^{\mathrm{target}}$ and dFIC is in equilibrium. For the examples in **Fig 4**, this set is a continuous line. However, the surface of attractors exhibits gaps in the interval $y_0^{\mathrm{d}} \in [0.02, 0.095]$. Hence, targets in this interval might lead to a corresponding gap in the set of ($I_{\mathrm{ext}}$, *wFIC*) values for which dFIC equilibria exist. In other words, dFIC can fail for targets in this range if $\mu$ is not chosen appropriately. Outside $y_0^{\mathrm{d}} \in [0.02, 0.095]$ on the other hand, dFIC will function as intended and control the system towards the desired target, independent of $I_{\mathrm{ext}}$. In summary, the correct application of the proposed homeodynamic inhibitory plasticity mechanism for a single JR node requires the existence of an attractor located at the desired $y_0^{\mathrm{target}}$ value (for LCs, this location is defined through their center), for a given constant $I_{\mathrm{ext}}$. The attractor does not need to be present at this location for the original JR model (*wFIC* = 1) but can be found through variation of *wFIC*.

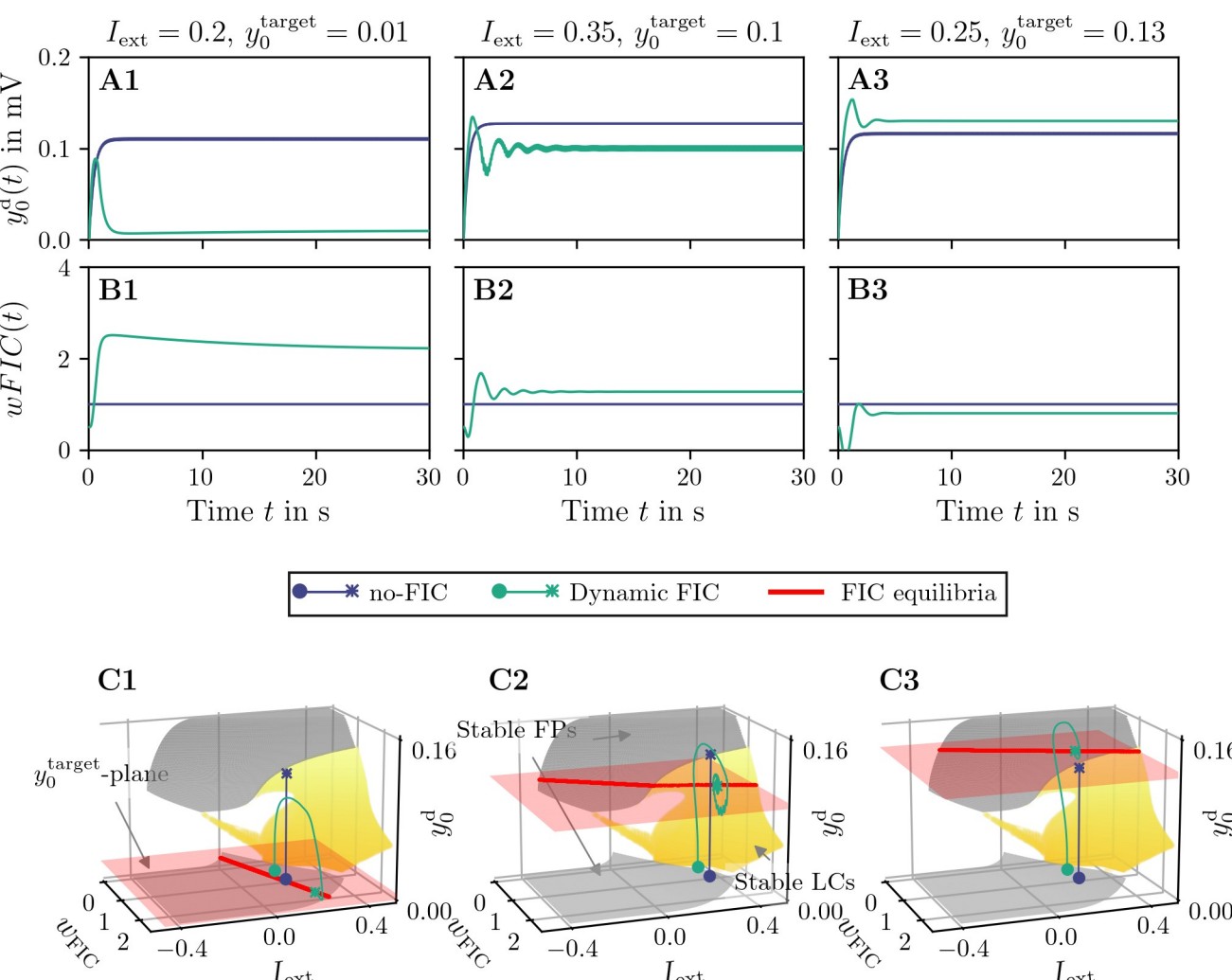

**Fig 4. Comparison of no-FIC ($\eta = 0$) and dynamic FIC ($\eta > 0$) of a single Jansen-Rit node. A, B:** $y_0^{\mathrm{d}}(t)$ and $wFIC(t)$ vs. time $t$, respectively, obtained by direct simulation of **Eqs (1), (2)** and **Eq (4)**. Each column corresponds to different values of $y_0^{\mathrm{target}}$ and $I_{\mathrm{ext}}$. **C:** Bifurcation structure of the uncoupled Jansen-Rit model with averaging in $w_{\mathrm{FIC}} - I_{\mathrm{ext}} - y_0^{\mathrm{d}}$ space. In C, $y_0^{\mathrm{target}}$ is marked by a red plane and the intersection of this plane with stable equilibria or limit cycles is shown as a red line and marks dFIC equilibria. Results obtained from direct simulation of the no-FIC ($\eta = 0$) and dFIC ($\eta > 0$) are superimposed. Circles mark the start and asterisks the end of these trajectories. The no-FIC and dFIC initial conditions are shifted slightly for visual clarity. The parameter values for the direct simulations are $\tau_d = 400$ ms and $\frac{1}{\eta} = 400$ ms $\times$ (mV)$^2$, other parameter values are given in **Table 1**. The bifurcation diagrams were computed numerically using AUTO-07p [60].

## dFIC in a brain network model

Moving on to the network case, $I_{\mathrm{ext},i}$ now contains the network input, is time-dependent and spans over a range of values, which is determined by the network activity and indegree $d_i = \sum_{j=1}^{N_{\mathrm{ROIs}}} C_{ij}$ of each node $i$. At the same time, $wFIC$ is now a vector with $N_{\mathrm{ROIs}}$ entries $wFIC_i$, one for each brain region. An analysis as above is considerably more intricate and beyond the scope of this work. Nevertheless, we want to point out how the above requirement for successful control through dFIC can be translated to a network setup, by making some assumptions. If the network connections are not strong enough to induce fast transients or overly synchronized oscillations, $I_{\mathrm{ext},i}$ can be separated into three components,

$$I_{\mathrm{ext},i}(t) = \mu + d_i R(t) + \xi_i(t), \tag{9}$$

namely the constant offset $\mu$, a slowly varying indegree-dependent mean-field input $d_iR$ and small fluctuations $\xi_i$, which arise from the network and are distinct to each node. If we neglect the last contribution, the constant $\mu$ remains and shifts the entire network towards larger external currents. The mean-field input $d_iR$ on the other hand introduces a spread of the network along $I_{\text{ext}}$, depending on the indegree distribution. Given that $R(t)$ is subject to only slow changes, the above slow-fast analysis for the single JR model holds valid, since $R$ can be treated in the limit case, just as *wFIC* before. Hence, the behavior of each node in the network can be approximated by the single-node bifurcation structure present in the plane ($I_{\text{ext}}$, *wFIC*). The major difference to the single-node case is the distribution of nodes along $I_{\text{ext}}$. As shown for the single node, outside the interval [0.02, 0.095] of the $y_0^{\text{target}}$ parameter, i.e., the gap in the attractor surfaces (**Figs 3B** and **4**), we find a continuum of $I_{\text{ext}}$ values, for which dFIC convergence is guaranteed. In the network case and with the assumption of weak coupling, this continuum of $I_{\text{ext}}$ will allow the individual node to approach the dFIC equilibrium, no matter its precise location on the $I_{\text{ext}}$-axis. Within the gap on the other hand, there can be nodes with $I_{\text{ext}}$ values for which no dFIC equilibrium exists. In other words, in the case of the JR model, dFIC equilibrium on the network level can fail for $y_0^{\text{target}} \in [0.02, 0.095]$.

In the upcoming sections, we will provide numerical evidence for the functioning of dFIC in different network setups. We assume the success criterion of tuning parameter convergence to be 1% of the distance of the mean of the last $L = 5$ seconds of the $y_0$ time-series from the $y_0^{\text{target}}$. Before convergence of *wFIC*$_i$ one can encounter damped oscillations towards the dFIC equilibrium. Hence, we choose initial conditions based on the bifurcation structure to reduce these transients and obtain faster convergence of *wFIC*$_i$. When the tuning was successful, we averaged *wFIC*$_i$ over last $L$ seconds to obtain a single *pFIC*$_i$ parameter per node in the brain network. We present the simulation results in two separate parts, one for a small network, used to explore the dFIC performance for different values of the time constant and learning rate parameters, $\tau_d$ and $\eta$, respectively, and then, for a biologically realistic whole-brain network.

## Small network simulations

The results of $\eta$ and $\tau_d$ parameter explorations demonstrate the convergence of the tuning algorithm (**Fig 5**) for $y_0^{\text{target}} = 0.01$ for all and $\tau_d$ values in the cases of $\eta = 0.01$ and 0.005. This means that all single-node *wFIC*$_i$ state variables converged asymptotically to a stable value necessary for the system to exhibit the desired activity level with the accuracy of 1%. The tuning to a $y_0^{\text{target}} = 0.103$ was successful across all conditions except for short $\tau_d = 10ms$ and a few $\eta$ values for $G = 10$, $G = 25$, and $G = 100$ as depicted on **Fig 5**. Based on these results, $\tau_d = 1000$, $\eta = 0.005$ were chosen as parameters for further for simulations as they proved to be robust across both $y_0^{\text{target}}$ and $G$ values. We have also tested a faster solution with $\tau_d = 100$, $\eta = 0.0025$, which, as expected, yielded almost identical results across both $y_0^{\text{target}}$ and all $G$ values. **Fig 6** demonstrates the resulting *pFIC*$_i$ values as functions of $G$ and indegree, showing a monotonously increasing—albeit nonlinear—relation between the respective values.

Afterwards, we run the post-FIC simulations to assess the impact of tuning in the small network. The time-series coming from simulations with $G = 1, 10, 25$ demonstrate the change in activity levels caused by the introduction of the pFIC parameter (**Fig 7**). It becomes evident that noise and network effects cause an offset between the long-term average activity level and the $y_0^{\text{target}}$ for two reasons: (a) predominantly—due to stochastic switches between attractors, which can lead to a net-positive (negative) effect for the sub-bistable (super-bistable) case (respectively), and (b) due to the (sigmoidal) nonlinearity of the model's activation function, which leads always to a net-positive offset, even in the absence of any switches. As a reminder,

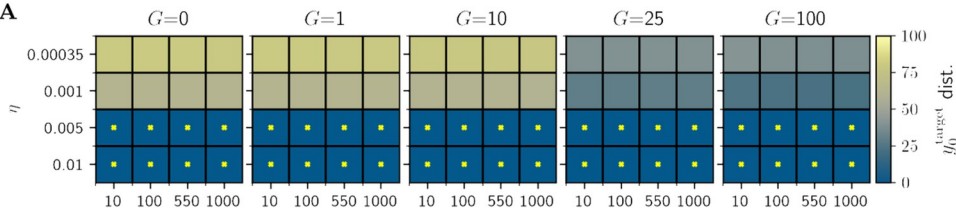

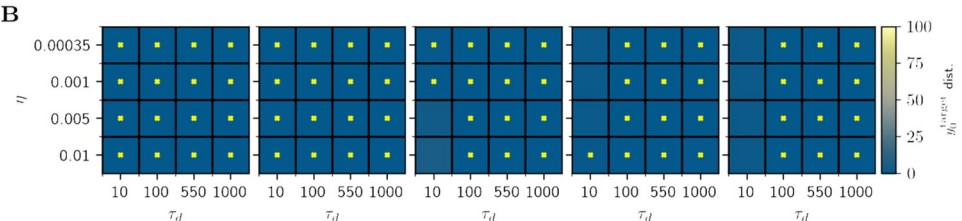

**Fig 5. Analysis of $\eta$ and time window ($\tau_d$) parameter explorations for $y_0^{\text{target}} = 0.01$ (A) and $y_0^{\text{target}} = 0.103$ (B) under different $G$ values (columns).** Successful tuning ($y_0$ mean within 1% of $y_0^{\text{target}}$) is marked by yellow crosses. The colors of the squares indicate the percentage of the distance of the y0 state variable activity from the $y_0^{\text{target}}$ value after tuning. In the two cases of $y_0^{\text{target}} = 0.01$ and $y_0^{\text{target}} = 0.103$, the evolution of the wFIC variable was on the correct trajectory but the tuning required a longer simulation to converge.

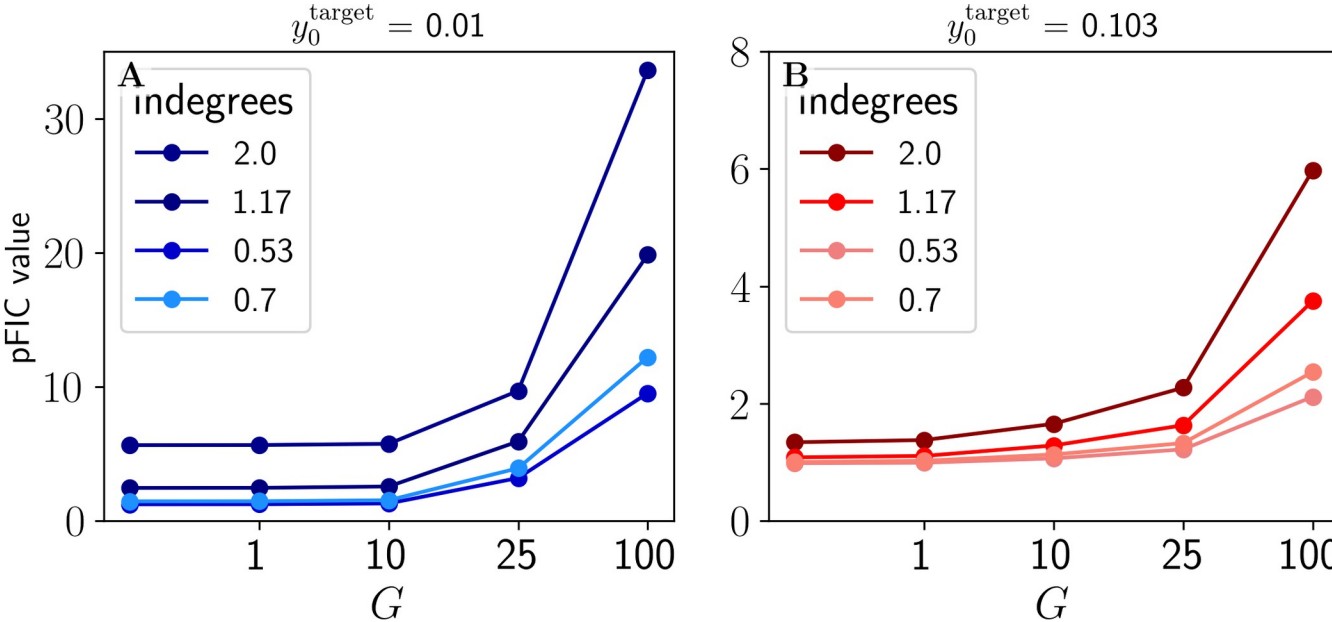

**Fig 6. Small-network differences in inhibitory control pFIC parameters per node obtained from the tuning procedure for different $y_0^{\text{target}}$ values (A: $y_0^{\text{target}}$ = 0.01, B: $y_0^{\text{target}}$ = 0.103) and $G$ values with the color intensity marking the indegree of a given node.** Plot shows the relationship between $G$ values, pFIC values with color-coded indegrees of the nodes of the small connectome. Two different target values were chosen to correspond to sub-bistable and super-bistable JR configurations. Due to the additive impact of noise in post-FIC simulations we chose the $y_0^{\text{target}}$ = 0.01 further to the left from the saddle node bifurcation and $y_0^{\text{target}}$ = 0.103 directly at the edge of saddle node bifurcation between the LCs (**Fig 2A**).

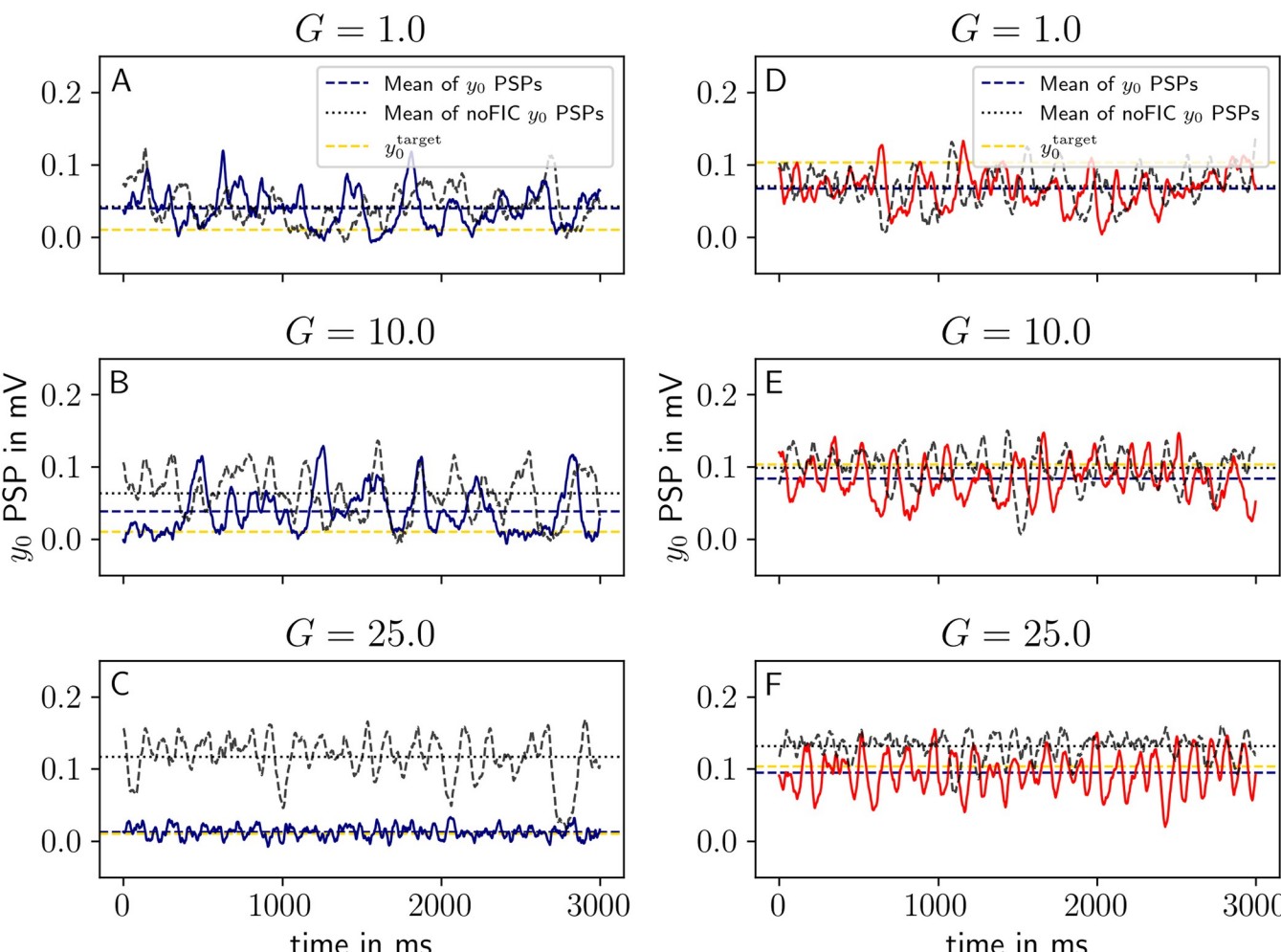

**Fig 7. Average post-FIC** $y0$ state variable for $y_0^{\text{target}} = 0.01$ in blue (panels **A-C**) and $y_0^{\text{target}} = 0.103$ (panels **D-F**) in red for different $G$ values compared against the no-FIC scenario in black for the small network PSPs exhibit bistability between the fixed point (FP) and limit cycles (LCs). The post-FIC potentials are plotted in blue/red solid lines and no-FIC potentials in black dashed lines. Effect of tuning is visible for $G > 1$. For $G = 10$, the deviation from the target is visible, which can be explained by the noise induced switching between two dynamical regimes of the system. Note that for $y_0^{\text{target}} = 0.01$ the $y_0$ average is higher than the target due to switches between FP and slow LC and for the $y_0^{\text{target}} = 0.103$ the effect is opposite, due to switches between dominant fast LC and slow LC.

here we consider sub-bistable targets to be $0.007 < y_0^{\text{target}} < 0.019$ and super-bistable $y_0^{\text{target}} \approx 0.1$.

## Whole-brain post-FIC results

In the second part of the results, we focus on the comparison between post-FIC and no-FIC simulations on the whole-brain network. The resulting PSP time-series show the desired effect of tuning, which, as expected, restricts the baseline activity of each of the nodes to the $y_0^{\text{target}}$ value. The difference between the $y_0^{\text{target}}$ and the mean post-FIC $y_0$ values is caused by the introduction of noise as well as the ability of noise to cause nodes to periodically jump between the regimes of the JR model. The time-series of the state variable $y_0$ show a predictable deviation from the target activity level, proportional to both the sum of weights and the chosen $G$ parameter value, further highlighting the fact that the dFIC for $y_0^{\text{target}} = 0.007$ and $y_0^{\text{target}} < 0.11$ places the system in the proximity of multi-stability. The switches between different regimes are less

visible for the respective no-FIC simulations (**S3 Fig**), a phenomenon further investigated in the next section.

We have computed and compared power spectra for the best fitting post-FIC simulations (using the Welch method), and best fitting no-FIC simulations. The power spectra analysis shows a significant post-FIC increase in power in the respective α and θ frequency bands when compared with the corresponding no-FIC simulations (**Fig 8**). Increases for $y_0^{\text{target}} = 0.007$ occur predominantly in the θ band as the tuning target places the system near the low FP (**Fig 8A**), where network fluctuations and noise move the system towards the bistable regime between the LCs (to the right on the bifurcation diagram). The presence of multiple peaks in the power spectra, corresponding mainly to θ and slow α rhythms (<*10* Hz), suggests that it is bistable switching and not the slow LC that explains the increased theta power. Similarly, the $y_0^{\text{target}} = 0.01$ is a critical point on the fast LC exhibiting alpha oscillations (**Fig 8B**), where changes in input and noise cause the periodical switches between both alpha and theta LCs. We conclude that the dFIC mechanism is responsible for the increase in power in the respective rhythms. Further support of this claim comes from the Poincaré map analysis.

## dFIC Improves the fit to empirical data

The results of the fitting based on $R_{FC}$ are shown in **Fig 9** and example matrices are available in the **S4 Fig**. The results indicate that dFIC improves fits between the simulated and empirical FC data as long as $y_0^{\text{target}} \leq 0.11$ allows the system to reach multiple dynamical regimes. When $y_0^{\text{target}} > 0.11$, all nodes are forced by the $pFIC_i$ parameter to remain on the fast LC and the repertoire of possible behaviors is significantly diminished. We assumed that the 30 min simulation for BOLD signals approximates well the stationary value of the FC statistic for a given parameter set (including the amplitude of noise).

FCD fitting results show an overall improvement of fit for most sub-bistable targets and significantly higher best similarity score ($1-KSD_{FCD}$) for post-FIC simulations compared with no-FIC settings (**Fig 10**).

We have combined the results from synchrony-adjusted FC and FCD fitting into a one Multimodal Factor fitness function (MMF). The resulting $y_0^{\text{target}}$ optima have largely remained similar, but the global optimum shifted to $y_0^{\text{target}} = 0.01$, $G = 11$ (**Fig 11**). Interestingly, the FCD distribution of this simulation visually fits the empirical FCD better than the best qualitative $1-KSD_{FCD}$ fit (**Fig 12B** compared with **Fig 12C**) as the best $1-KSD_{FCD}$ only fit contained a second peak.

The plot of all the MMF fits in **Fig 13** shows the wide range of good fits for a wide range of sub-bistable $y_0^{\text{target}} \in [0.007, 0.0189]$ and *G* values and for super-bistable $y_0^{\text{target}} = 0.1$. The difference between the wider sub-bistable $y_0^{\text{target}}$ range and a single super-bistable value can be attributed to two factors: first, the denser sampling of the former parameter space and the noise, since the offset it induces via network effects is mainly net-positive for the sub-bistable regime due to switches from the FP to the LC.

To test if differential MMF fitting results between the post-FIC and no-FIC simulations are statistically significant, we conducted a permutation statistical analysis by sampling without replacement from the same set of MMF values computed for the above shorter time windows, but this time randomly mixing the two conditions of post-FIC and no-FIC simulations. Statistical analysis showed that the difference between post-FIC and no-FIC simulations was significant at a level of $p < 10^{-6}$ for almost all parameter combinations. Details of both procedures are further described in the **S1 Appendix**.

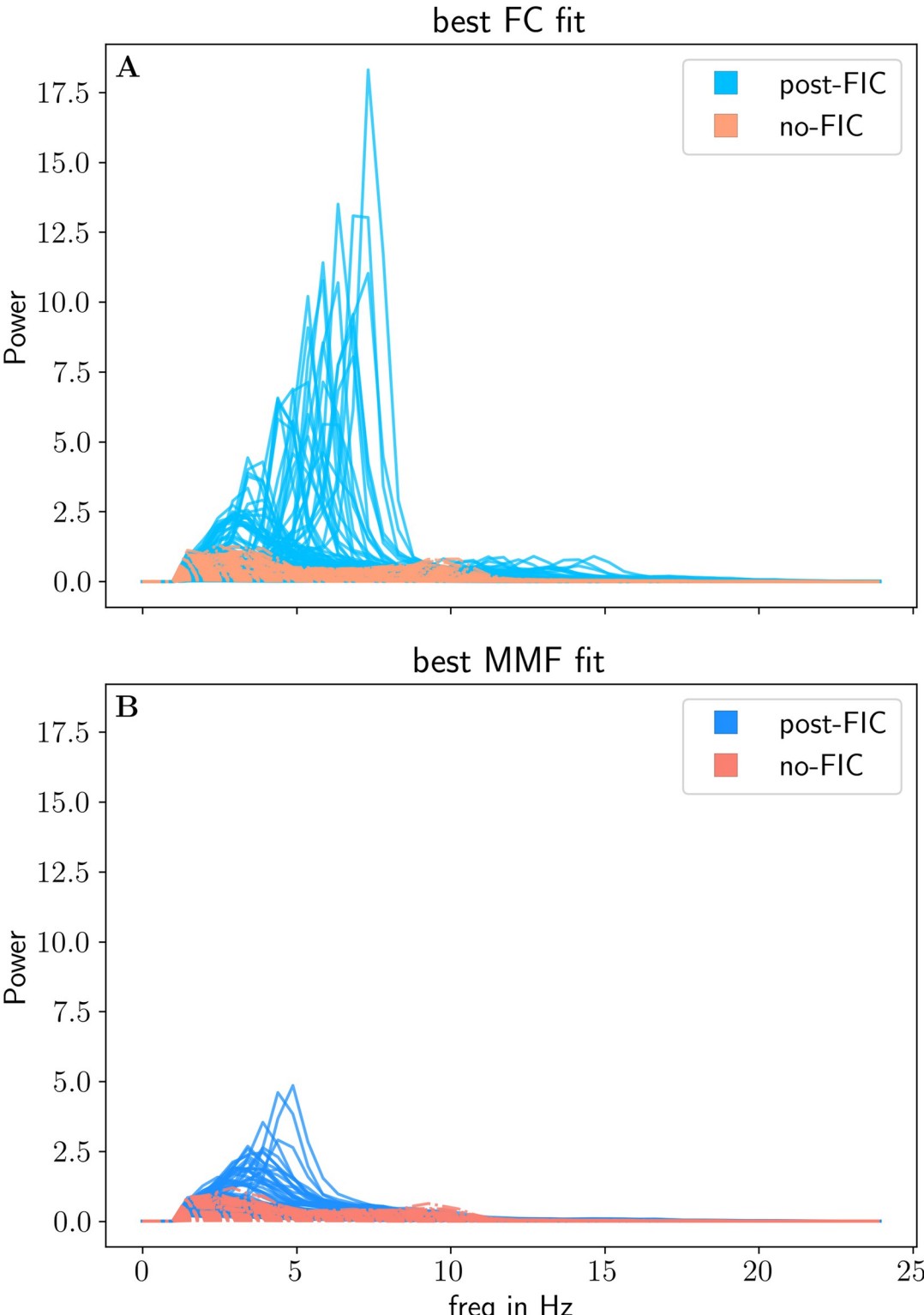

**Fig 8. The power spectra of all nodes for the two best post-FIC simulations for two different fit measures $y_0^{\text{target}}$ values (panel A: FC-fit and panel B: MMF fit, blue) and two best fitting no-FIC simulations (red).**

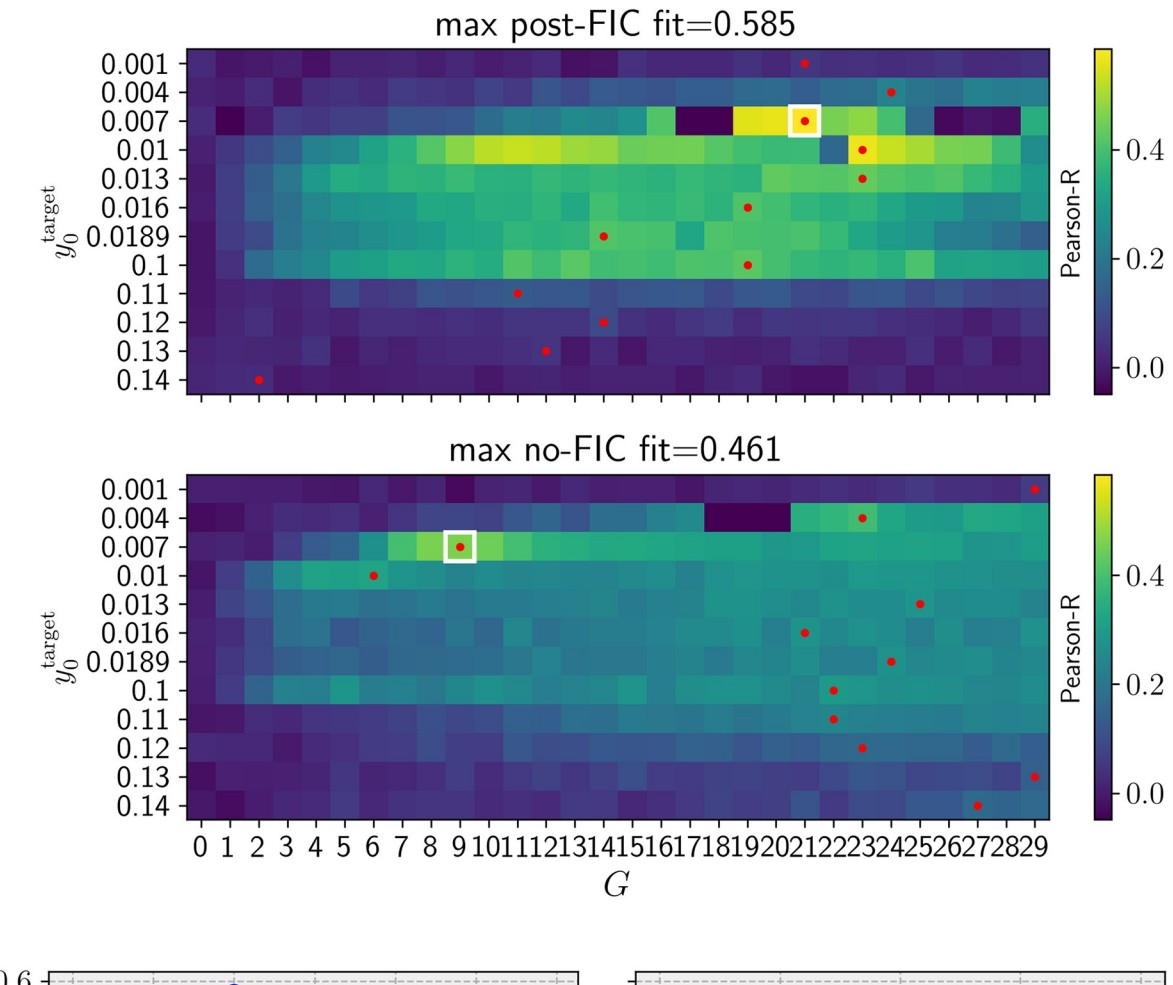

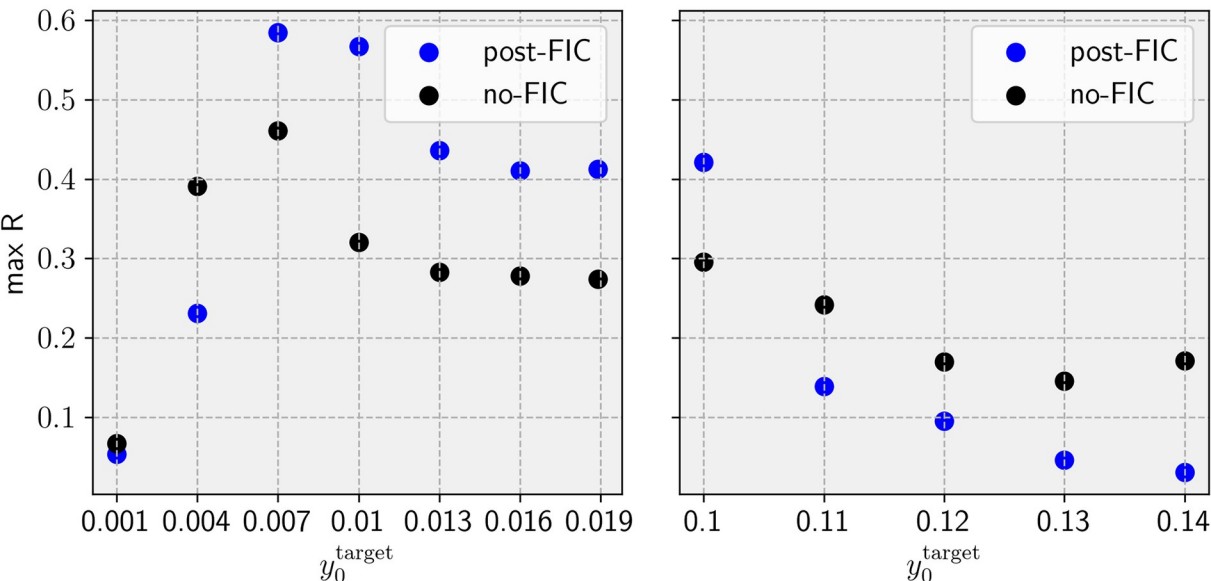

**Fig 9. Results of functional connectivity fitting. A-B.** The heatmap with the resulting fit measure for all post-FIC and the best fitting simulations without FIC (no-FIC). The red dots mark the best-fitting $G$ parameter per $y_0^{\text{target}}$, white square marks the best fitting simulation overall. **C-D.** Plots **C-D** illustrate the direct comparison of Pearson-R values for best fitting simulation per $y_0^{\text{target}}$ **C**: for sub-bistable and **D** for super-bistable target values. Max R.: synchronization-adjusted Person correlation coefficient between simulated and empirical data.

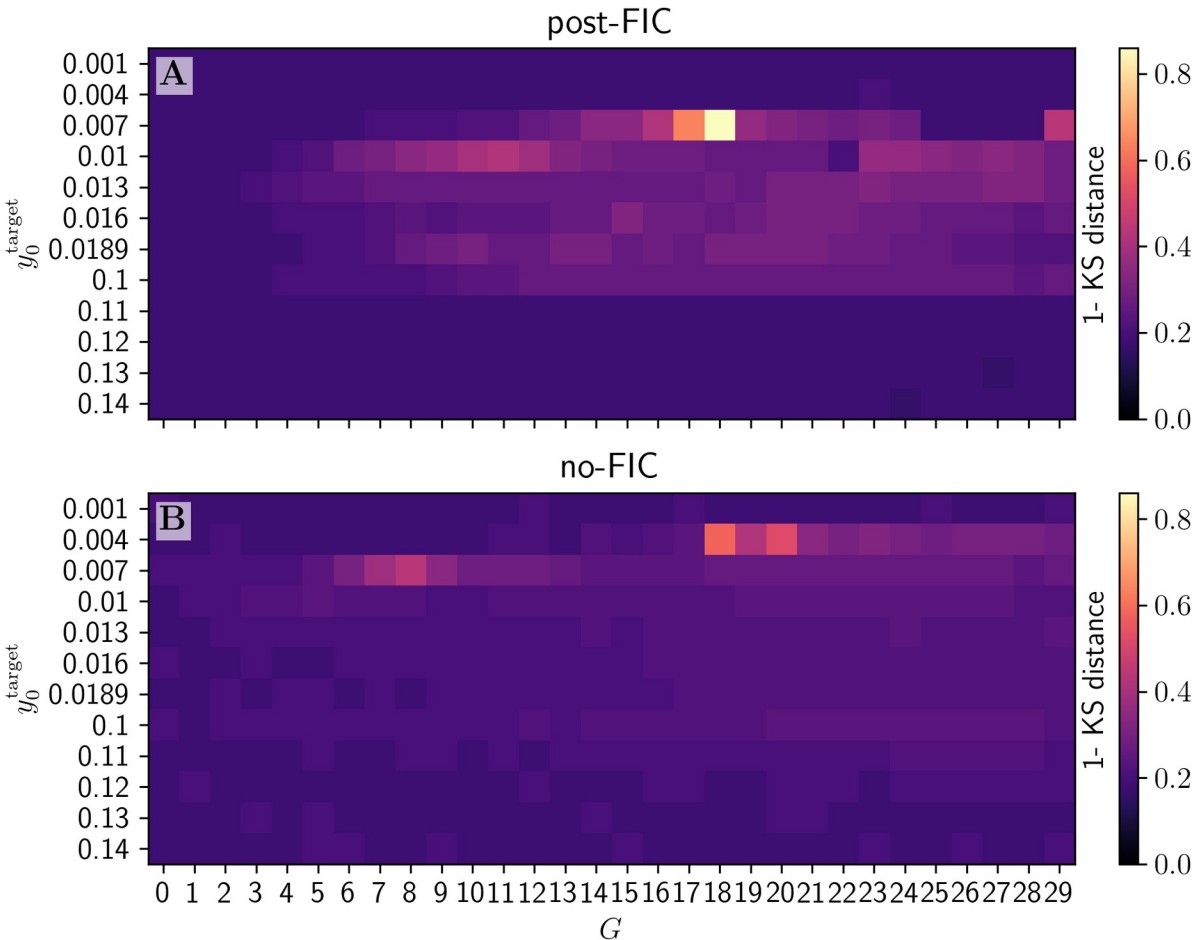

**Fig 10. Heatmap of Kolmogorov-Smirnov-Distance-based similarity measure (1-KSD) for both post-FIC (A) and no-FIC (B) simulations.** The optima are in a very similar range, but they differ slightly suggesting that this measure captures different (dynamical) properties of the BOLD dynamics.

### dFIC intensifies "switching" between dynamical regimes

To investigate the switching behavior of the model, we computed Poincaré maps of synaptic potential time-series. This step allowed us to highlight and quantify the effect of the dFIC on the model's underlying activity. First, we have identified a threshold of activity ($c = 6$ mV) which can be applied to the resulting maps to identify the regime of the JR model being exhibited by any node at any given time. More in detail, the lower activity FP of JR in presence of noise, yields points ($PSP_{i,k}$, $PSP_{i,k+1}$) in [0 mV, 6 mV]×[0 mV, 6 mV], the fast LC in [6 mV, 18 mV]×[6 mV, 18 mV], whereas one period of the slow LC will have one point in [0 mV, 6 mV]×[6 mV, 18 mV] and one in [6 mV, 0 mV]×[0 mV, 6 mV], due to the formation of two maxima within one period. If a node traverses multiple regimes however, the point cloud of a single node is going to fulfill multiple of those conditions, allowing to identify which regimes were visited and therefore which dynamics exhibited. By analysis of the pattern formed by individual nodes over time and the pattern of all nodes together, it is possible to extend the understanding of the overall behavior of the model. Further we have calculated how many nodes exhibit behaviors in one, two or three different regimes for more than 7,5% of the simulation time $T_p$, as a measure of complexity of the resulting signals.

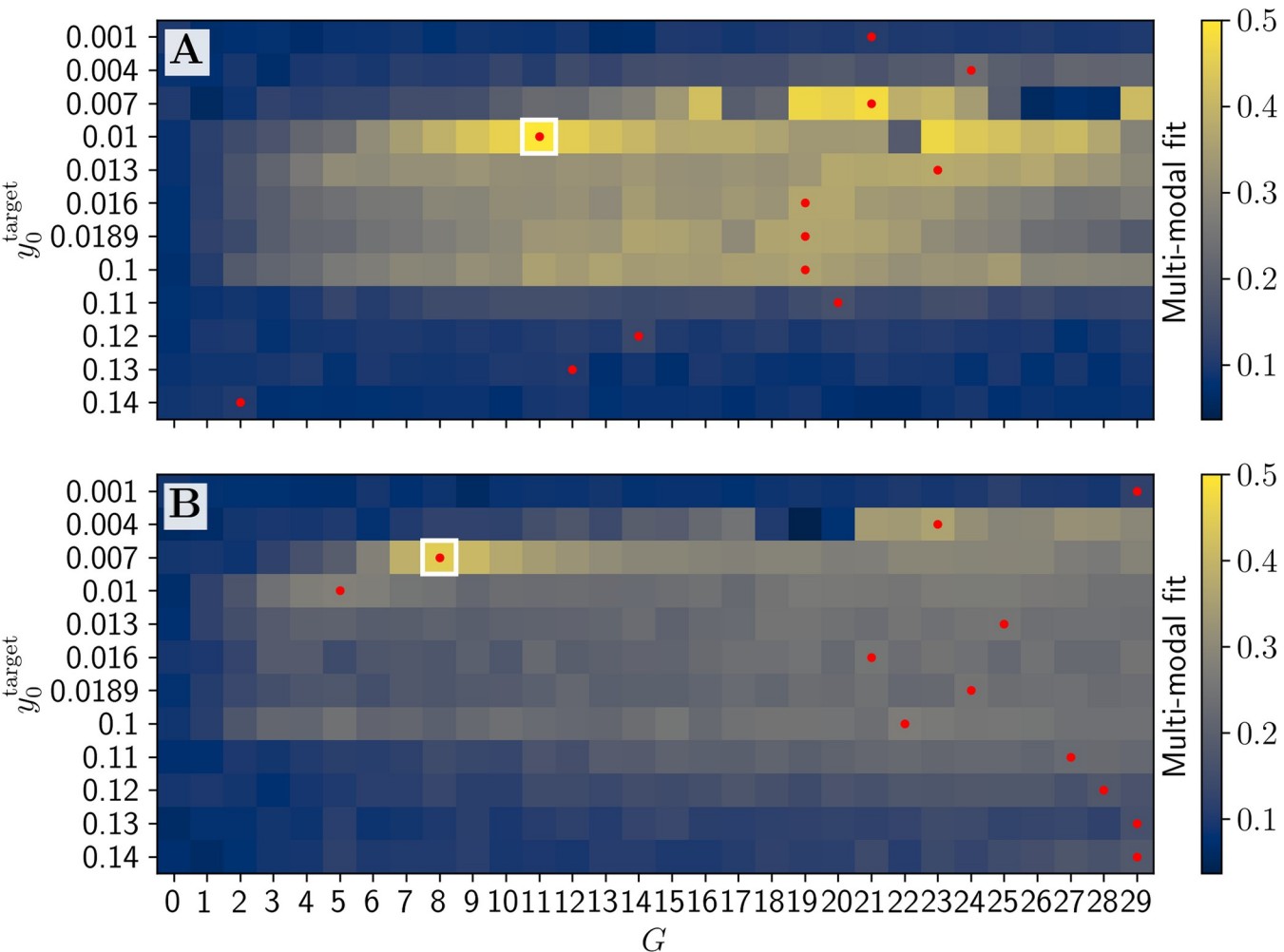

**Fig 11. Heatmap of Multimodal Factor fit based for both post-FIC (top) and no-FIC (bottom) simulations.** The result that MMF maximizes not directly at the critical point of saddle-node bifurcation ($y_0^{\text{target}} \approx 0.0189$) but at $y_0^{\text{target}} = 0.01$ can be explained by two factors: (1) we sampled the $y_0^{\text{target}}$ space more densely for the sub-bistable regimes than for the super-bistable and (2) ones, as we suspected this range to be optimal, second: the network effects and noise have primarily additive effect, which causes a shift in effective network average of $I_{\text{ext,i}}$ to the right of the $y_0^{\text{target}}$ value.

For most of targets in the proximity to critical points of the JR model ($0.007 \leq y_0^{\text{target}} \leq 0.1$), Poincaré analysis shows higher variability of exhibited regimes. That means, more regimes can be reached by any single node and a higher number of nodes are 'visiting' a higher number of regimes in post-FIC simulations compared to their no-FIC counterparts. The dFIC algorithm for those targets increases in overall switching behavior for a high number of tuning targets. In contrast for some low $G$ values and $y_0^{\text{target}} \in \{0.004, 0.007, 0.01\}$ dFIC is binding the activity of the nodes to a single regime. Expectedly, for $y_0^{\text{target}} \leq 0.004$ and $y_0^{\text{target}} > 0.11$, dFIC does not significantly impact the switching behavior. **Fig 14** shows the $y_0^{\text{target}}$ and $G$ values for which FIC caused an increase in the number of showcased behaviors. We demonstrate this effect based on two examples: **Fig 15** shows best fitting no-FIC ($y_0^{\text{target}} = 0.007, G = 9$) compared with best fitting post-FIC ($y_0^{\text{target}} = 0.01, G = 11$), where tuning increases this number significantly (**Fig 15A1 and 15A2**). This effect can also be observed when the $y_0^{\text{target}}$ is super-bistable, in which case the no-FIC method is compared with the post-FIC one, both with $y_0^{\text{target}} = 0.1$ and $G = 11$ (**Fig 15B1 and 15B2**).

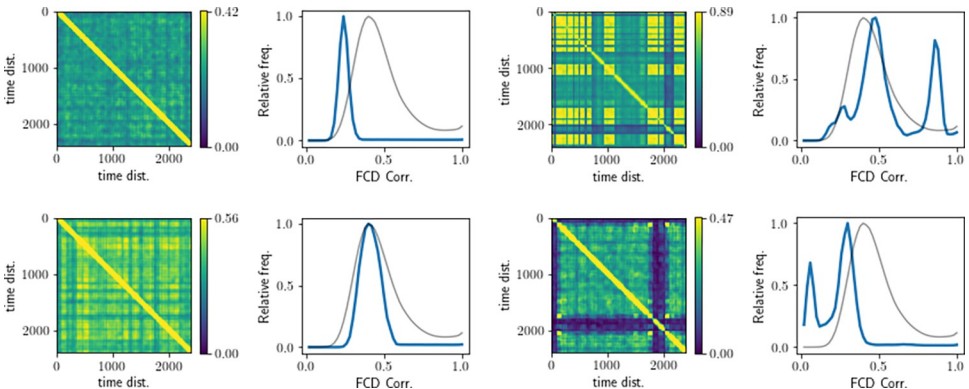

**Fig 12. Example of FCD matrices and their distributions. A**: shows the FCD data for the best fitting post-FIC simulation chosen solely based on the Pearson correlation with empirical FC. **B**: for the generic empirical FCD data, **C**: for the combined MMF fit, **D**. shows the FCD data for the best fitting no-FIC simulation based on the combined $1 - KSD_{FCD}$ fit. On all the plots average empirical FCD distribution is plotted in black. The FCD matrices were threshold at $96^{th}$ percentile for clarity.

Overall, Poincaré mapping analysis for the best fitting simulations according to the MMF showed the positive impact of dFIC on the number of nodes traversing 2 and 3 regimes (**Fig 14**). The overlap between high values of fits of $R_{FC}$, $1-KSD_{FCD}$, and MMF in the $(y_0^{\text{target}}, G)$ parameter plane and the results of our Poincaré mapping analysis suggest a strong link between the overall quality of fits between the simulated and empirical BOLD data and the switching behavior in the PSP time-series. This effect is visible while comparing the heatmaps (**Fig 16**), suggesting a similar range of potentially optimal JR and dFIC parameters. Our results provide further evidence that in simulations with JR model the switches between different regimes leading to different amplitudes and frequencies of oscillations in the underlying PSP

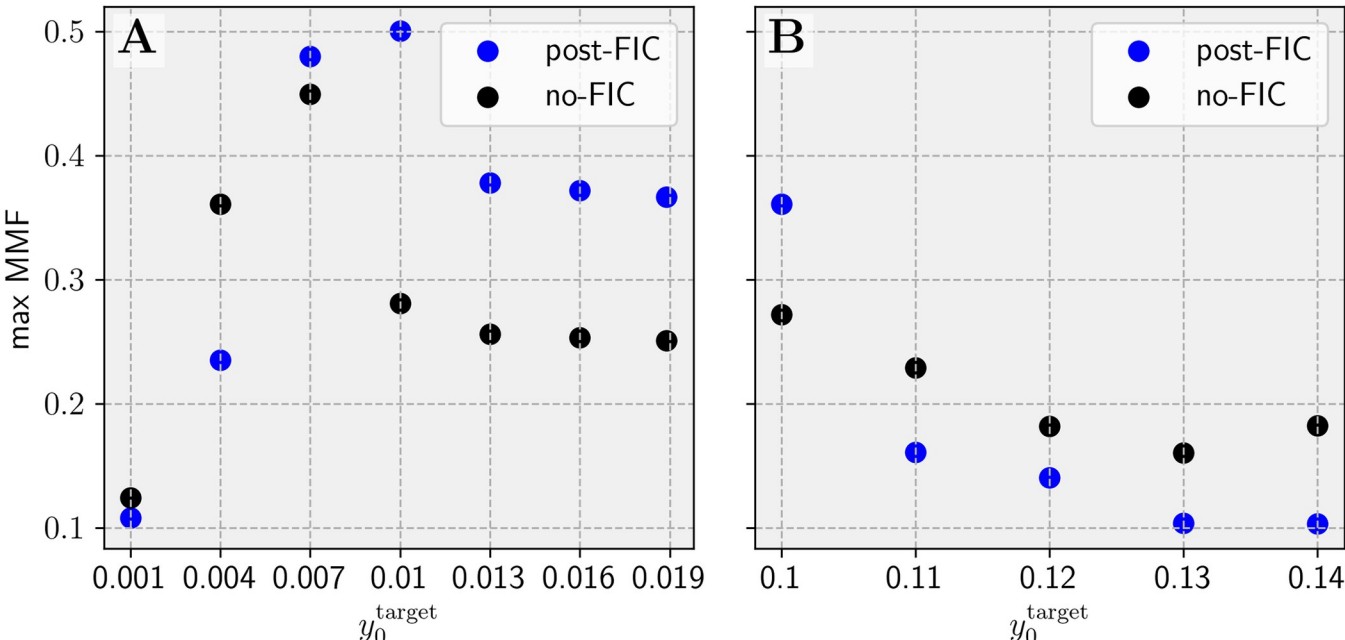

**Fig 13. Comparison of best fitting results for different $y_0^{\text{targets}}$. A**: Sub-bistable targets. **B**: Super-bistable targets. These plots show the viability of the Multimodal Factor (MMF) fitting approach and the positive effect of the dFIC on the range of sub-bistable cases and directly super-bistable case of $y_0^{\text{target}} = 0.1$.

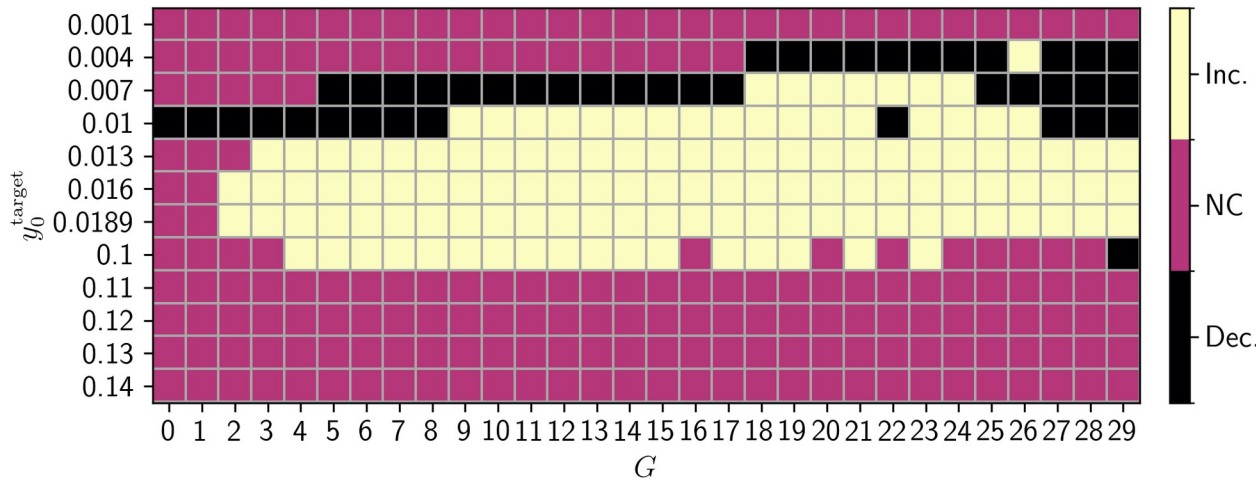

**Fig 14. Summary heatmap of increases in number of regimes traversed by the nodes.** If the number of regimes of post-FIC simulations is greater than for no-FIC simulations the map is marked in yellow, in the opposite case in black, and in yellow if there is no difference. The tuning targets that correspond to a single attractor, with large distance to any bistability show lack of improvement with dFIC or decrease in number of nodes in multiple exhibited regimes. In contrast the targets where the system is tuned to a sub-bistable regime and directly super-bistable regime show increases in that measure. Inc: increased, Dec: decreased, NC: no change.

data result in more variable and hence more biologically plausible BOLD signals and BOLD-derived FC matrices.

## Discussion

The work presented in this paper addresses issues of over-excitation and large divergence in individual node activity arising from coupling multiple nodes of neural mass models in a whole-brain network with heavy-tailed weight distributions (39,41,57). The goal was to provide a solution that (i) does not involve changing the underlying structural connectivity (ii) allows for decoupling the effects of global coupling on correlations among regions and excitability for each single region, (iii) permits direct analysis of adaptation dynamics. Established methods of counteracting this issue include: transformation of the connectome weights (i.e. log-scale transformations) [73,74], fitting a $G$ parameter to reduce the relative strength of all network effects on a single node, and various implementations of inhibitory plasticity mechanisms [16,46,50,51]. Since those methods did not fulfill all three of those conditions, we proposed dynamic Feedback Inhibition Control (dFIC) as a solution. We introduced a previously established FIC mechanism [16,50,57,59] into the differential equations of the model and provided an analytical treatment of the adaptation procedure. We determined the conditions under which dFIC control is successful in the JR model, as well as the practical implications of using dFIC in terms of fitting data and the dynamical repertoire of the regimes of the model.

We analyzed the case of an isolated node during tuning and established limits for tuning targets. To the best of our knowledge, this was the first attempt at providing analytical results on the functioning of a homeodynamic synaptic plasticity mechanism in a BNM. We demonstrated that—for a wide range of $G$ values—we can tune the values of the $wFIC_i$ such that the deterministic system is kept at a predetermined $y_0^{target}$ value. The presence of a stable attractor associated with the target constitutes the main constraint of the $y_0^{target}$ selection. Using small network simulations, we have studied the interplay between $\tau_d$, $\eta$, and $wFIC_i$ to converge and performed small network post-FIC simulations to assess the effects of applying dFIC in a network. Further, we have analyzed the effects of dFIC in a whole-brain BNM setup and

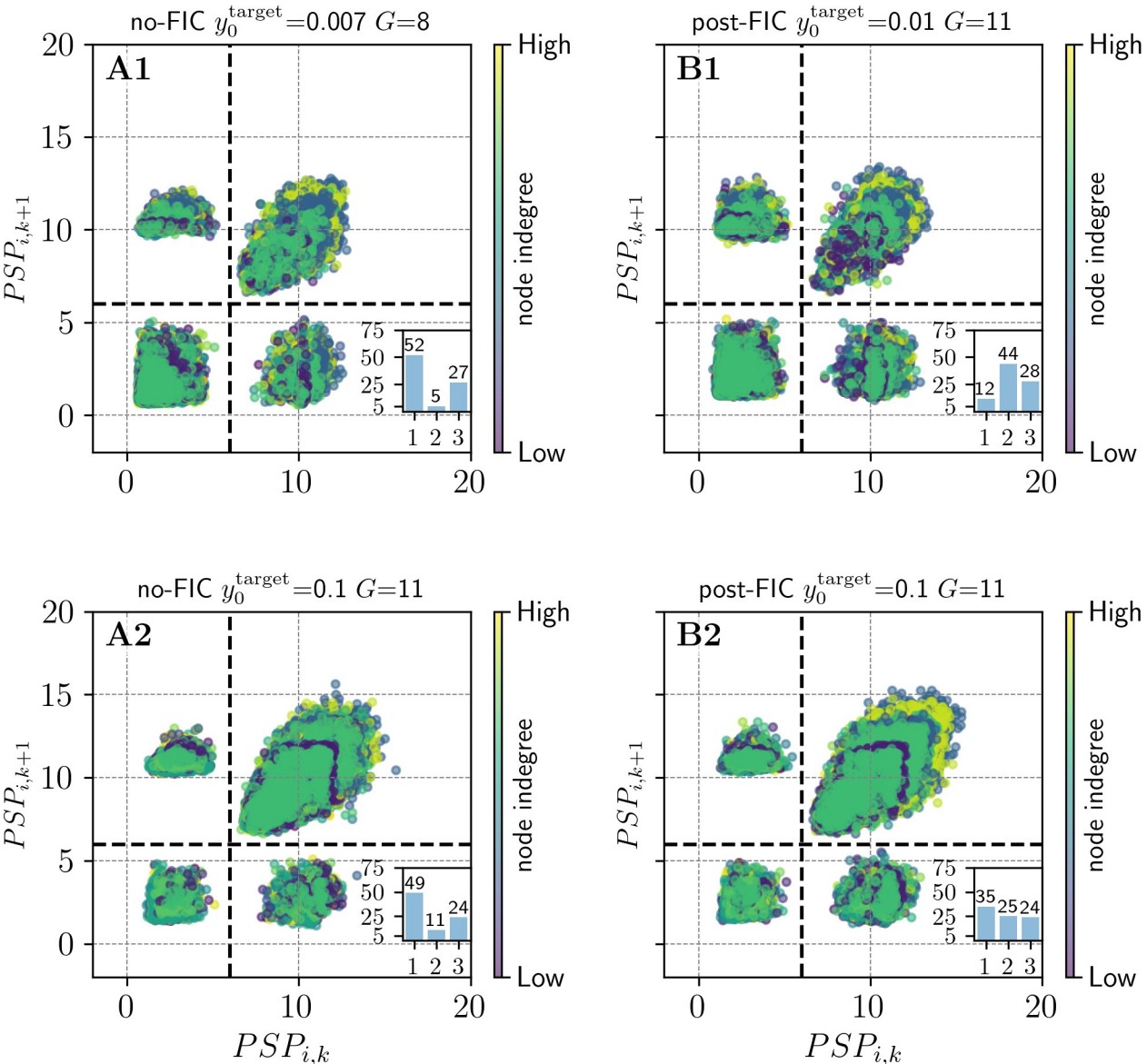

**Fig 15. Poincaré maps for the best no-FIC (A1) and post-FIC (B1) MMF fit.** Each point ($PSP_{i,k}$, $PSP_{i,k+1}$) maps one PSP maximum $PSP_{i,k}$ of an individual node i to the next one $PSP_{i,k+1}$. The node indegrees are color-coded (low indegree: dark blue, high indegree: yellow). The set threshold at c = 6 mV consistently (visually) separates the FPs and LCs of the JR model. Bar plots at the bottom right corner of each subplot show the percentage of nodes exhibiting 1,2 or 3 regimes during at least 7,5% of the time of the simulation, showing the effect of dFIC. The **A2** and **B2** panels compare no-FIC with post-FIC simulations for super-bistable settings ($y_0^{\text{target}} = 0.1$, G = 11). The node indegrees are color-coded low indegree: dark blue, high indegree: yellow. The set threshold at c = 6 mV consistently (visually) separates the FPs and LCs of the JR model. Both plots demonstrate a difference in number of regimes traversed by each (map) of the nodes during the simulation. A decrease in number of nodes in a single regime or an increase in the number of nodes in two or three regimes is considered a richer repertoire of behaviors as a result of implementing dFIC.

compared the simulation results with the original JR model, including PSP data analysis and both static and dynamic BOLD time-series analysis.

There are several important outtakes from our results. First, our work shows that dFIC can be used to regulate the dynamical regimes of individual nodes in the whole-brain network. Based on the single-node bifurcation diagram of the model (**Fig 1**), one can choose a $y_0^{\text{target}}$ to determine which regimes of the model will be traversed when the system is coupled, and noise

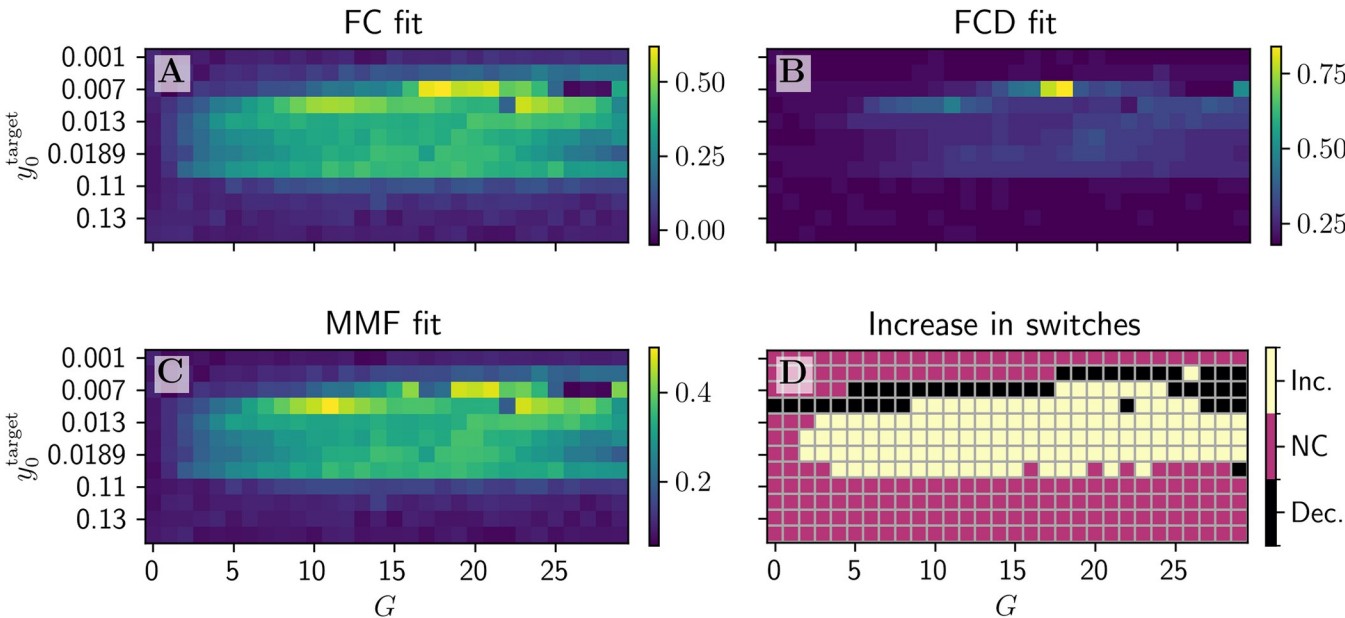

**Fig 16. The heatmaps showing FC, FCD, MMF fits and the heatmap of increase switching behavior in underlying PSP activity as shown by Poincare analysis.** Heatmaps for all 3 fitting measures (**A**-**C**) show a similar range of good fit parameter values spanning from $y_0^{\text{target}} = 0.007$ to 0.1. and for wide range of $G$ values. Interestingly a very similar range shows an increased number of switches during post-FIC simulations in PSPs (**D**).

added. The noise levels need to be within certain limits to be investigated either computationally or via an analysis of the resulting stochastic system. Too low noise will likely not lead to any switches between attractors, whereas too high noise will lead to dynamics hardly determined by the specific structure of the system. In network simulation results, we have observed three outcomes of dFIC depending on the target: (i) dynamics of a node reflect a regime at set target, (ii) dynamics correspond to a regime at the set target with a small positive offset, and (iii) dynamics correspond to a regime at the set target with a small negative offset. The offsets can be attributed to network effects, noise, and nonlinearities. In the JR model, this feature allows for the selection of the frequencies of interest, and control over the overall variability in the system, which was not the case in known-to-us implementations of JR model coupled in a BNM. As an example, if $y_0^{\text{target}}$ corresponds to an oscillatory regime, dFIC results in the increase in power in the associated dominant $\alpha$ and $\theta$ frequencies in obtained signals (compared to the established JR implementations [52,54–56].

Second, BOLD signals generated in presence of dFIC capture characteristics of empirical data better over a broad parameter region than without dFIC. The complexity of the PSP time-series, assessed through Poincaré maps, increases in presence of dFIC: the individual nodes of the BNM traverse more dynamical regimes. In particular, our findings show that when the tuning target is set near criticality (i.e. either in proximity to a saddle node bifurcation $0.007 < y_0^{\text{target}} < 0.189$ or at the bistable regime between the two LCs $y_0^{\text{target}} \approx 0.1$), an enriched landscape of exhibited regimes results together with an increased frequency of transitions between them compared to both: the best fitting simulations with dFIC for other $y_0^{\text{target}}$ and the best fitting JR simulations without the dFIC. This increased complexity is associated with a better fit between simulated and empirical data as we show in **Fig 16** also when dynamic FC and synchronization are taken into consideration. Predictably, tuning the system to high activity $y_0^{\text{target}} > 0.11$ values, yielded worse fits than the respective no-FIC simulations, due to all

regions being constrained to the fast LC, providing additional argument for the role of dynamical variability in BOLD fitting.

Lastly, by exploring several tuning targets and $G$ values, we have showcased our solution's ability to decouple the $G$ parameter's influence on inter-regional correlations from the overall excitability level, in line with the FIC tuning results by Deco et al. [16]. This has an interesting consequence, as one can *a priori* decide on the activity target of interest, and, thus, determine the range of available dynamical regimes of the system. This constitutes a more informed starting point for the exploration of the effect of other model parameters or of stimulation.

## dFIC, E-I balance and criticality

The balance of excitation and inhibition, postulated as a central property of a healthy neural system, has had a few explicit implementations in research dedicated to brain simulations [8,46,51,75]. Studies show the significance of homeodynamic plasticity in the context of maintaining E-I balance and simulating healthy and more realistic neuronal dynamics, but the connection to the notion of criticality remains debated [25,76,77]. Both concepts have been argued to require some sort of plasticity mechanism [33,36,49,78,79]. It is worth noting that several mechanisms collectively contribute to the brain's ability to self-regulate E-I balance through synaptic plasticity [80,81]. Recent studies suggest the roles of short-term [82] and long-term synaptic plasticity [83], and structural plasticity [84] in maintaining this balance. Additionally, there are cellular mechanisms distinct from plasticity, but with capability of self-regulation. As an example, spike-frequency adaption, i.e., the property of certain types neurons to reduce their firing frequency upon sustained input, could effectively act as a negative feedback similar to feedback inhibition [85]. Overall, mechanisms like dFIC are implementations of generalised inhibitory synaptic plasticity mechanisms in the BNMs. However, their effectiveness in maintaining E-I balance in simulated networks does not allow for determining the specific biological underpinnings of such mechanisms.

Even though, the E-I balance does not necessarily entail criticality in the context of BNM [86], there has been evidence suggesting that E-I balance, and optimal information processing based on criticality are two tightly related phenomena [24,31,87,88]. For example, researchers have postulated that E-I balance is directly responsible for bringing the cortical activity to criticality in a rat's visual cortex *in vivo* [89]. Their research strongly suggests that homeostatic plasticity, linked to E-I balance in general, is responsible for bringing the cortex to the critical state. A recent review on inhibitory plasticity mechanisms points out that E-I balance in neural networks enables complex, non-linear collective behavior [75]. Furthermore, E-I balance can be a necessary condition for more complex dynamics on a whole network scale. Proponents of E-I balance have argued for the role of inhibitory plasticity mechanism in keeping the BNM at criticality in the Deco-Wang model and Wilson-Cowan model, respectively [18,49]. They have provided evidence for the role of such a process suggesting that the brain uses such mechanisms to stay in a state that prevents runaway activity such as epileptic seizures and enriches the dynamics (thanks to proximity to criticality), while staying relatively immune to the strength of noise or global coupling. Interestingly, their results, similar to ours, showed improved correspondence between simulated and empirical functional resting-state fMRI connectivity compared to models without inhibitory plasticity mechanism.

Our findings support the idea that E-I balance plays an important role in maintaining rich and biologically plausible dynamics in BNMs. Depending on the $y_0^{\text{target}}$, dFIC positively affects the desired characteristics of simulated neural signals such as variability and richness of behavior. Here, we have provided experimental arguments for choosing the sub-bistable or directly super-bistable targets in the context of the JR model. However, our solution delineated limits

and conditions for a wider range of tuning targets including the low-activity or high-activity states, which in future research can be valid tuning targets depending on the specific research question.

The mechanisms related to inhibitory control, similar to dFIC, have potential relevance to modeling E-I balance also in the clinical context. Parameters responsible for the balance between excitation and inhibition have been linked to abnormalities in Alzheimer's Disease, such as synaptic degeneration, hyperexcitation, white matter atrophy, as well as amyloid-beta and tau levels. In previous studies, inhibitory parameters of BNM were fit to empirical functional data of patients [9,90]. Recently, the JR model has been used to derive biomarkers related to inhibitory control, emphasizing the potential of model-based approaches in the early diagnosis of dementia and Alzheimer's disease [57,91]. Similarly, the role of inhibitory control in the progression of dementia has been studied using the Wilson-Cowan model [92] and the Deco's model from 2018 [93,94]. As we have shown, dFIC improves the overall model fitting, facilitating the study of how different dynamical regimes and transitions between them may be linked to various pathological conditions. Additionally, the resulting inhibitory plasticity parameters, can be interpreted as measures of dynamical differences in clinical setting.

To sum up, dFIC is designed to maintain the activity of all nodes at a certain level, by adjusting local inhibition and, in that manner, maintain E-I balance at both local and—indirectly—at the global scale regardless of the chosen target. Our exploration of different activity targets suggests that tuning the system to the proximity of bistability results in increased switching between regimes and better fits to the empirical data. dFIC provides improvement in characteristics observed in the empirical signals. Previous works have treated FIC as an iterative process of simulating, computing activity levels, adjusting weights and then simulating again [8,16,17]. In this work, we embed dynamic FIC as an extension into the JR model, similar to Santos et al. in Wilson-Cowan model [51]. The dFIC equations lead to a slow, asymptotic convergence towards a target average activity of the JR nodes in the whole-brain network model, under conditions relatively easy to be met in most cases (of neural mass models and target dynamical regimes thereof). The analytical treatment of the single-node tuning dynamics reveals the conditions under which convergence towards a target average activity can be guaranteed and provides guidelines for employing dFIC in practice.

## Limitations

We have implemented the dFIC mechanism in the JR model, because it is widely used in BNM studies as well as because it is complex enough to allow for multiple dynamical regimes and possible bifurcations among them. However, our results are applicable to other models which have a control quantity (here $y_0^d$) which monotonously decreases with an increasing control parameter (here *wFIC*). One can assume that a dFIC mechanism, potentially implemented in other neural mass models, would possess this property, since increased inhibition should naturally lead to reduced activity. Beyond this, the stability of the dFIC dynamics only depends on the presence of stable fixed points and LCs (in the uncontrolled model), which are located at the desired target activity. Further analysis is needed to assess how dFIC would change a system's behavior which possesses chaotic attractors. Therefore, applying dFIC to other neural mass models should be straightforward, a bifurcation analysis being a prerequisite. Future work can take up this task and provide examples of dFIC application to other neural mass models implemented in TVB and frequently used in BNM studies.

In general, although our results assume deterministic dynamics during the dFIC tuning process, we expect them to remain valid in the presence of relatively low additive, white noise, which is generally used in stochastic simulations of BNMs, as well as for our post-FIC

simulations. Such a stochastic version of dFIC could also resolve the problem of a possible off-set between the $y_0^{\text{target}}$ and the average $y_0$ value resulting from stochastic post-FIC simulations, which was observed in our study. Alternatively, tuning in presence of noise could be achieved by running a short stochastic dFIC simulation right after the deterministic dFIC has converged, by using the result of the latter as the initial condition of the former.

Our work utilized fitting static and dynamic characteristics of the simulated BOLD signals to assess their 'biological plausibility', including FC, FCD and synchrony. However, it did not extend to the high-temporal resolution signals such as MEG and EEG. Even though, we presented the basic properties of the generated PSP time-series (with and without dFIC), in the frequency domain and via Poincare Maps (**Figs 8,14 and 15**), we did not extend our fitting procedures to include those other signal modalities. Such multimodal fitting can help in the future determine the relationship between different modalities and further improve both the fitting methods and the homeodynamic fitting procedures.

Further research can also investigate dFIC as a homeodynamic mechanism, possibly relying on inhibitory plasticity, which maintains the state of the brain within limits and responds to external perturbations induced by e.g., cognitive stimuli. This kind of slow timescale modulation of brain dynamics could be of great interest in understanding data, e.g., on FCD dynamics of BOLD signals. In this context it is important to note that we based our simulation on average empirical structural and functional data: average SC, average FC, and average distribution of FCDs were derived from a large HCP dataset. Hence, when considering data of individual subjects, dFIC might yield slightly different results. The overall findings suggesting sub-bistable and directly super-bistable targets as optimal should in the future be tested using the individual subject data, both for normative and clinical cases.

## Conclusion

We demonstrated how dFIC can be used to regulate network activity at the single-node level, stabilizing it at a predetermined target value. The analytical treatment of the single node tuning allowed us to determine the ranges of possible activity-level targets. The comparison of post-FIC simulation results with the JR model without such mechanism, both shows the practical value of using the proposed mechanism and provides supporting arguments for extending similar solutions to other models used for whole-brain simulations. We intended dFIC to be a useful addition to the widely used JR model, which can improve the control over networks of JR nodes by addressing the issue of over-excitation. We kept the proposed solution flexible–and easy to extend to other neural mass models—such that future users can test it further and decide how to set up a BNM depending on their own research goals, and independent of their decisions regarding the changes to the SC, or the intended effect of the global coupling.

## Supporting information

**S1 Fig. Example Poincaré map of the PSP from a single simulation.** Each point ($PSP_{i,k}$, $PSP_{i,k+1}$) maps one PSP maximum $PSP_{i,k}$ of an individual node $i$ to the next one $PSP_{i,k+1}$. The node indegrees are color-coded low indegree: dark blue, high indegree: yellow. The set threshold at c = 6 mV consistently (visually) separates the FPs and LCs of the JR model.
(TIF)

**S2 Fig. Schematic of the dynamic Feedback Inhibition Control procedure.** Steps: (i) $\mu$ and $y_0^{\text{target}}$ selection, (ii) deterministic tuning resulting in wFIC$_i$ vector (which needs to be checked for convergence), (iii) the last 5 seconds of the vector need to be averaged per node to obtain

pFIC$_i$ parameter, (iv) stochastic (optional) post-FIC simulation, (viii) simulation output.
(TIF)

**S3 Fig. Comparison of the mean of whole-brain time-series with dFIC (post-FIC) vs. without dFIC (no-FIC).** For **A**: $y_0^{\text{target}} = 0.01$ and **B**: $y_0^{\text{target}} = 0.1$. In both cases, the difference in the baseline of the displayed timeseries and illustrates the desired effect of tuning: In case **A** the dFIC limits the network-induced increase in activity despite the addition of the noise, preventing the nodes from exhibiting fast limit cycle oscillations. In case **B**, dFIC places the nodes at the upper boundary of bistability, allowing the system to exhibit not only fast limit cycle oscillations but also slow limit cycle oscillations–due to noise and network effects.
(TIF)

**S4 Fig. Graphical representation of the best fitting functional connectivity (FC) matrices.** Based on pure FC correlation (top-left) and best fitting simulation after correcting for synchrony (top-right), best fitting FC matrix based on MMF measure (bottom-left) and average empirical FC (bottom-right). empFC: empirical functional connectivity; FC: functional connectivity; MMF: multi-modal factor.
(TIF)

**S1 Appendix. Statistical analyses.**
(PDF)

# Acknowledgments

Data were provided by the Human Connectome Project, WU-Minn Consortium (Principal Investigators: David Van Essen and Kamil Ugurbil; 1U54MH091657) funded by the 16 NIH Institutes and Centers that support the NIH Blueprint for Neuroscience Research; and by the McDonnell Center for Systems Neuroscience at Washington.

# Author Contributions

**Conceptualization:** Jan Stasinski, Michael Schirner, Dionysios Perdikis.

**Formal analysis:** Halgurd Taher, Dionysios Perdikis.

**Funding acquisition:** Petra Ritter.

**Investigation:** Jan Stasinski.

**Methodology:** Jan Stasinski, Halgurd Taher, Dionysios Perdikis.

**Project administration:** Jan Stasinski, Petra Ritter.

**Resources:** Petra Ritter.

**Software:** Jan Stasinski.

**Supervision:** Halgurd Taher, Jil Mona Meier, Michael Schirner, Dionysios Perdikis, Petra Ritter.

**Visualization:** Jan Stasinski, Halgurd Taher, Jil Mona Meier, Petra Ritter.

**Writing – original draft:** Jan Stasinski, Halgurd Taher, Dionysios Perdikis.

**Writing – review & editing:** Jan Stasinski, Halgurd Taher, Jil Mona Meier, Michael Schirner, Dionysios Perdikis, Petra Ritter.

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
