## [Decision Letter · Decision Letter 0]

30 Jul 2024

Dear Mr Stasinski,

Thank you very much for submitting your manuscript "Homeodynamic feedback inhibition control in whole-brain simulations" for consideration at PLOS Computational Biology.

As with all papers reviewed by the journal, your manuscript was reviewed by members of the editorial board and by several independent reviewers. In light of the reviews (below this email), we would like to invite the resubmission of a significantly-revised version that takes into account the reviewers' comments.

We cannot make any decision about publication until we have seen the revised manuscript and your response to the reviewers' comments. Your revised manuscript is also likely to be sent to reviewers for further evaluation.

Sincerely,

Marcus Kaiser, Ph.D.

Academic Editor

PLOS Computational Biology

Lyle Graham

Section Editor

PLOS Computational Biology

Reviewer's Responses to Questions

**Comments to the Authors:**

Reviewer #1: This paper presents an analysis of a recently developed method for the stabilisation of brain dynamics in large-scale neural mass model simulations. The method is based on the idea that activity homeostasis can be achieved by adjusting inhibitory weights in an activity-dependent manner to maintain a set-point firing rate. The key insight used here is that global stability is maintained when this set-point corresponds to an attractor of a single-node model (the Jansen-Rit model in this paper).

The authors show that this method can control the target activity level to obtain biologically plausible neural activity, including central features of BOLD signals. Interestingly, the method is also shown to maintain to high variability of model dynamics corresponding to critical/bifurcation points hypothesised to be important for brain function. Although presented mainly as a technical solution to stabilise large-scale brain simulations, the work also adds support to the hypothesis that neural homeostasis through inhibitory plasticity may be functionally important - I wonder if this idea could be explored in a data-driven way with the methods presented here.

I have no major comments on the paper, in my opinion it is well written and argued. I think though that equations 1-4 miss the dynamics (it looks like eqn. 4c should be the derivative of wFIC). Second, it would be prudent to document the published code to enable re-use. As it is, the repository only has some python files without any documentation or instructions.

Reviewer #2: This article suggested the methodology to obtain biologically plausible simulation of a brain by improving widely used Jansen-Rit model, and performed thorough analyses about how the authors’ method could regenerate the characteristics of brain signal, especially the dynamical features of common complex system and neuroscientific metrics. Although the authors handled mean-field model and BOLD data proficiently and mostly convincing results, I would like to recommend this article for publishing only after significant efforts would be made for improving the quality of manuscript.

1. Overall readability should be significantly improved.

1-1) There is substantial number of honest mistakes across the whole paper. For instance, reference 18, 19, 59, 72, 84, 92 were obviously broken. Equation 6 was not indexed. At line number 485, referring figure number went wrong. I could find a dozen more even after initial superficial reading of the manuscript, and it is hard to list all that I found while repetitive readings. It could be simply minor points to improve but many mistakes arouse a doubt that there might be overlooked critical ones. I would like to recommend that the authors thoroughly check especially, figure, equation, reference number in the text.

1-2) Several acronyms were not introduced properly. Ex) JR at line number 152, FC at 164.

1-3) All the coefficients and variable names should be at least once briefly explained. Ex) T in eq 5a, R_FC in eq 6.

1-4) Table 1 was never referred in the text. In addition, though JR model and its coefficient set are well-known, the supporting prior works or reasoning to determine the values should be offered in the text.

1-5) ‘Introduction’, especially line 74-126 or line 127-188 can be substantially shrunk by removing redundant messages and rearrangement to ‘discussion’. The reason why I brought up this issue is that because of this long introduction compared to the very specific and neat motivation and summary of the work, the main novelty gets blurred, resulting in unnecessary complexity of the flow.

2. Since the authors emphasized that the introduction of dFIC into JR model improves the pipeline to construct a better BMN regenerating ‘biologically plausible’ brain signal, the novelty of this work was spotlighted mainly in the perspective on ‘observation-simulation matching.’ Although JR model and the other types of mean-field models don’t target the biological plausibility (or similarity) of local network architecture, dFIC seems capable of somewhat compensating such simplification. I would like to recommend for the authors to state this point briefly. Although somehow, in Discussion, the section stating E-I balance is weakly related to this point, it seems possible that a bolder interpretation, such as dFIC can be the biological mechanism of local E-I balance, can be suggested.

3. As the extension of 2, it would be much better to be discussed, the determination of wFIC and pFIC in the pathological conditions (one of the main purpose of TVB) and the implications of resultant wFIC and pFIC.

4. Poincare map

4-1) In Figure 2, I did not understand how Poincare map without varying lag time or transition map (between dots) could specify the dynamics of each regime such as slow limit cycle or fast limit cycle.

4-2) In the caption of fig2, ‘each color marks a single node’ and ‘colors on the map mark the indegree of the nodes’ are very confusing comments.

4-3) As an extension, it would be better to kindly offer the full details of the Poincare map analysis in Method section.

4-4) In Figure 15 in the subplots, traversing number 3 seems not changing in any case (~25). I would like to recommend that the authors offer the interpretation on it.

Reviewer #3: In this manuscript, Stasinki et al. proposed a new version of the Jansen & Rit model which includes a Feedback Inhibitory Control (FIC) mechanism, similar to the one implemented in Abeysuriya et al. (Plos Comp Biol, 2018) and Deco et al. (Journal of Neuroscience, 2014). They first introduced the model, and then explored how the FIC can change the dynamics at the single region level through bifurcation analyses. The authors performed small network simulations to show the capability of the FIC mechanisms to preserve E/I balance in network simulations when increasing the global coupling. Finally, they characterized the model’s performance in fitting empirical fMRI BOLD functional connectivity and functional connectivity dynamics empirical data.

I found the article very interesting to read, and methodologically robust. There’s no doubt that the authors performed not only exhaustive but also correct analyses to demonstrate the plausibility and usefulness of incorporating FIC into the Jansen & Rit model. The figures are of high quality, and the results completely support the main hypotheses. In the field of whole-brain modeling, an implementation of FIC mechanisms (or other plasticity-based mechanisms for preserving E/I balance) is needed in the field of neural mass modeling using the Jansen & Rit model. I would recommend this manuscript if the authors can address the following major and minor suggestions.

Majors

The article would benefit from a reduction of content, especially in the Introduction section, which is quite long. For example, the paragraph that starts in line 97 “from a dynamical systems perspective…” is redundant compared to the one before it. The two paragraphs of lines 114 and 127 are both about E/I balance, hyperexcitability, and FIC, and can be reduced and combined into one. The paragraph of line 143 can be reduced or removed. Other phrases, like “This version of FIC is formulated as a set of ordinary differential equations…” (line 180) can be moved to Methods, and other ones like “The analytical treatment of the single-node tuning dynamics reveals the conditions…” (line 185) may be more properly placed in the Results.

What is the motivation of the authors for using three differential equations for including the FIC mechanisms, instead of just one (like in Abeysuriya et al., Plos Comp Biol, 2018)? For me, it wasn’t clear when reading the text. What is the advantage of including the additional two state variables in the model? Perhaps the authors can address this point in the Discussion, and/or compare the model’s outputs or performance with a version of the model that includes only the equation 4c.

The authors discarded simulated FCs with a mean > 0.25. This can be avoided using metrics that “balance” connectivity pattern similarity and differences in global correlations, like the structural similarity index (like in Ipiña et al., 2020, Neuroimage).

Why the authors didn’t use the SSIM or other composed metrics (similar to Pfeffer et al. 2021, Science Advances)?

The authors showed that the model can fit fMRI BOLD FC and FCD. However, it would be good to show that the model is also able to fit some properties of empirical EEG (e.g., EEG FC, power spectrum). The authors can try to fit both fMRI and EEG dynamics using the model and the data from the HCP. If not, it would be nice at least to see what the EEG power spectrum looks like when using the optimal parameters of the fMRI fitting. I also recommend addressing this limitation of the work in the Discussion if not addressed.

In the Introduction section, Line 152, the authors said that “… so far, an FIC mechanism is missing for the Jansen & Rit model”. Indeed, a mechanism for preserving the E/I balance in the Jansen & Rit model was proposed by Coronel-Oliveros et al. (2024, Alzheimer's & Dementia). I think that the authors should mention this paper as an antecedent of FIC mechanisms in the Jansen & Rit model. The mechanism proposed in Coronel-Oliveros et al. is very similar (but of course not equal) to the implementation proposed by Stasinki et al. here, without considering the additional activity detection equations (4a and 4b). Perhaps the authors can discuss the differences between these two versions of the Jansen & Rit model.

It would be a nice addition to the Discussion to talk about possible clinical applications, in a very short paragraph, of the JR + FIC model, for example, proposing how the FIC can be used to study E/I disbalances in dementia (see for example Ranasinghe et al., 2022, eLife; Martinez-Cañada et al., 2023, Alzheimer’s & Dementia: Diagnosis, Assessment and Disease Monitoring; Coronel-Oliveros et al., 2024; Alzheimer’s and Dementia; Moguilner et al., 2024; Alzheimer’s Research & Therapy). In this other work, the authors used the JR model for proposing biomarkers from the model, and not from the data: Amato et al., 2024, Alzheimer’s & Dementia: Diagnosis, Assessment and Disease Monitoring).

Minors

Line 26 in Abstract: “Using a variety of metrics”. What metrics?

Line 30-31 in Abstract: the two lines might be reduced to “… of fitting simulated to fMRI static and dynamic functional connectivity data”.

The Figure 1 description (and the figure itself) should be moved to the Results section.

It would be nice to include a scheme or drawing of the model. Perhaps the authors can move the Fig S1 in Supp Material to the Main Text.

Line 346 in results: “Before convergence of wFIC_i one can encounter…” This may be moved to the Results section.

Line 391: “We have identified…” - move to Results.

Figure 2 should be in Results, not Methods.

Line 398 Fitting methods: how many model realizations (random seeds) did you use for stochastic simulations?

Line 413 in Methods: “Into a single multi-modal fitness …” add the acronym (MMF).

Line 429 in Methods: “To test if differential MMF fitting…”. That entire part may be moved to Results.

Line 538 in Results: “In summary, the correct application…”. That phrase is redundant and can be removed.

Line 814 in Discussion: “In contrast, when the model was tuned to…”. Limit to discuss the results, not repeat them in the Discussion.

Line 823 in Discussion: “Overall, dFIC gives more…”. I think this might be removed.

Line 834 in Discussion: “The criticality hypothesis says that…”. This was explained in the Introduction, and the Discussion section

**Have the authors made all data and (if applicable) computational code underlying the findings in their manuscript fully available?**

Reviewer #1: Yes

Reviewer #2: Yes

Reviewer #3: Yes

PLOS authors have the option to publish the peer review history of their article (what does this mean?). If published, this will include your full peer review and any attached files.

Reviewer #1: **Yes: **Matthias H Hennig

Reviewer #2: No

Reviewer #3: **Yes: **Vicente Medel
---

## [Decision Letter · Decision Letter 1]

25 Oct 2024

Dear Mr Stasinski,

We are pleased to inform you that your manuscript 'Homeodynamic feedback inhibition control in whole-brain simulations' has been provisionally accepted for publication in PLOS Computational Biology.

Best regards,

Marcus Kaiser, Ph.D.

Academic Editor

PLOS Computational Biology

Lyle Graham

Section Editor

PLOS Computational Biology

Feilim Mac Gabhann

Editor-in-Chief

PLOS Computational Biology

Jason Papin

Editor-in-Chief

PLOS Computational Biology

Reviewer's Responses to Questions

**Comments to the Authors:**

Reviewer #1: A good and comprehensive revision.

Reviewer #2: The authors carefully addressed my questions and suggestions for correction and reinforcement of the manuscript, and the revised manuscript is adequately improved. Especially, the current version delivers their message in focused manner with full information about the simulation that they performed. Thus, I would like to recommend this version of manuscript to be published.

Reviewer #3: The authors have succesfully answered all my inquiries.

**Have the authors made all data and (if applicable) computational code underlying the findings in their manuscript fully available?**

Reviewer #1: None

Reviewer #2: Yes

Reviewer #3: Yes

PLOS authors have the option to publish the peer review history of their article (what does this mean?). If published, this will include your full peer review and any attached files.

Reviewer #1: No

Reviewer #2: **Yes: **Taegon Kim

Reviewer #3: **Yes: **Vicente Medel

---

## [Editor Report · Acceptance letter]

19 Nov 2024

PCOMPBIOL-D-24-00577R1 

Homeodynamic feedback inhibition control in whole-brain simulations

Dear Dr Stasinski,

I am pleased to inform you that your manuscript has been formally accepted for publication in PLOS Computational Biology. Your manuscript is now with our production department and you will be notified of the publication date in due course.

With kind regards,

Lilla Horvath
